# A human ESC-based screen identifies a role for the translated lncRNA *LINC00261* in pancreatic endocrine differentiation

Bjoern Gaertner[1†], Sebastiaan van Heesch[2†‡], Valentin Schneider-Lunitz[2], Jana Felicitas Schulz[2], Franziska Witte[2], Susanne Blachut[2], Steven Nguyen[1], Regina Wong[1], Ileana Matta[1], Norbert Hübner[2,3,4,5], Maike Sander[1]*

[1]Departments of Pediatrics and Cellular & Molecular Medicine, Pediatric Diabetes Research Center, University of California, San Diego, San Diego, United States; [2]Cardiovascular and Metabolic Sciences, Max Delbrück Center for Molecular Medicine in the Helmholtz Association (MDC), Berlin, Germany; [3]DZHK (German Centre for Cardiovascular Research), Partner Site Berlin, Berlin, Germany; [4]Berlin Institute of Health (BIH), Berlin, Germany; [5]Charité -Universitätsmedizin, Berlin, Germany

*For correspondence:
masander@ucsd.edu

[†]These authors contributed equally to this work

Present address: [‡]The Princess Máxima Center for Pediatric Oncology, Utrecht, Netherlands

Competing interests: The authors declare that no competing interests exist.

Reviewing editor: Lori Sussel,

**Abstract** Long noncoding RNAs (lncRNAs) are a heterogenous group of RNAs, which can encode small proteins. The extent to which developmentally regulated lncRNAs are translated and whether the produced microproteins are relevant for human development is unknown. Using a human embryonic stem cell (hESC)-based pancreatic differentiation system, we show that many lncRNAs in direct vicinity of lineage-determining transcription factors (TFs) are dynamically regulated, predominantly cytosolic, and highly translated. We genetically ablated ten such lncRNAs, most of them translated, and found that nine are dispensable for pancreatic endocrine cell development. However, deletion of *LINC00261* diminishes insulin[+] cells, in a manner independent of the nearby TF *FOXA2*. One-by-one disruption of each of *LINC00261*'s open reading frames suggests that the RNA, rather than the produced microproteins, is required for endocrine development. Our work highlights extensive translation of lncRNAs during hESC pancreatic differentiation and provides a blueprint for dissection of their coding and noncoding roles.

## Introduction

Defects in pancreatic endocrine cell development confer increased diabetes risk later in life (*Bakhti et al., 2019*). Therefore, a detailed understanding of the factors that orchestrate endocrine cell differentiation is highly relevant to human disease. Many of the molecular mechanisms that underlie the formation of pancreatic endocrine cells have been defined (*Romer and Sussel, 2015*; *Schiesser and Wells, 2014*). However, despite some evidence that long noncoding RNAs (lncRNAs) are important for proper development and function of pancreatic beta cells (*Arnes et al., 2016*; *Morán et al., 2012*; *Wong et al., 2019*), a systematic functional assessment of the noncoding transcriptome during pancreas development is lacking.

Most lncRNAs with to date demonstrated roles in the regulation of fundamental developmental processes are active in the cell's nucleus (*Daneshvar et al., 2016*; *Jiang et al., 2015*; *Klattenhoff et al., 2013*; *Kurian et al., 2015*; *Lin et al., 2014*; *Luo et al., 2016*; *Ramos et al., 2015*). However, a large proportion of lncRNAs is predominantly cytosolic (*Cabili et al., 2015*; *van Heesch et al., 2014*), and the functional relevance of these lncRNAs has remained unexplored in the context of human development. It is now widely accepted that many cytosolic lncRNAs possess short, 'non-canonical' open reading frames (sORFs) that are translated (*Bazzini et al., 2014*;

*Makarewich and Olson, 2017*; *Ruiz-Orera et al., 2014*). What fraction of these non-canonical ORFs is functional, and whether sORF translation serves a pure regulatory purpose or results in the production of stable microproteins, remains an active topic of debate (*Levy, 2019*; *Ruiz-Orera et al., 2018*). Since high rates of conservation have historically been employed for the identification and annotation of canonical protein coding sequences (*Lin et al., 2011*; *Mudge et al., 2019*), a primary reason for doubting the protein-coding capacity of sORFs in presumed lncRNAs is their generally poor sequence conservation across species. To address these questions, several recent studies have systematically assessed the biological activity of newly discovered sORFs, revealing that many produce evolutionary young microproteins with roles across cellular organelles and processes, and a subset being essential for cell survival (*Chen et al., 2020*; *Martinez et al., 2020*; *Prensner et al., 2020*; *van Heesch et al., 2019*). This previously unrecognized coding capacity of supposedly noncoding RNAs illustrates their functional diversity and has called into question the noncoding classification of some lncRNAs. Thus, there is a need for careful investigation and dissection of any gene's coding and noncoding functions.

LncRNAs, translated or fully noncoding, are not randomly distributed in the genome but are frequently located close to, and coregulated with, canonical protein-coding genes in cis (*Luo et al., 2016*; *Neumann et al., 2018*; *van Heesch et al., 2019*). For example, the lncRNAs *DIGIT* (also known as *GSC-DT*) and *Gata6as* (also known as *lncGata6* or *GATA6-AS1*) have been reported to enhance expression of divergently expressed endoderm regulators *Goosecoid* (*GSC*) and *Gata6*, respectively (*Daneshvar et al., 2016*; *Luo et al., 2016*; *Neumann et al., 2018*). Similarly, the *Pax6*-associated lncRNA *Paupar* promotes pancreatic islet alpha cell formation through the alternative splicing of *Pax6* transcripts in mice (*Singer et al., 2019*). Furthermore, *LINC00261* (also known as *DEANR1*) and its neighboring TF *FOXA2* are both induced in endoderm formation, during which *LINC00261* has been proposed to positively regulate *FOXA2* expression (*Jiang et al., 2015*). However, whether such *cis*-acting lncRNAs are translated and may exert cytosolic functions through *trans*-acting, microprotein-dependent mechanisms relevant for endoderm and pancreas development is not known.

In this study, we classified lncRNAs based on their dynamic regulation, subcellular localization, and translation in a hESC differentiation system that recapitulates in vivo pancreas development. Next, we used this classification to prioritize select dynamically regulated and highly translated lncRNAs for deletion in hESCs, followed by extensive phenotypic characterization across multiple intermediate states of pancreas development. Nine out of the ten selected lncRNAs were not essential for pancreatic development and, despite their vicinity to lineage-determining TFs, none of these lncRNAs regulated the expression of these TFs in cis.

The deletion of one lncRNA, *LINC00261*, impaired human endocrine cell development and led to a significant reduction in the number of insulin-producing cells. Contrary to previous studies of *LINC00261* knockdown hESCs (*Jiang et al., 2015*), deletion of *LINC00261* had no effect on the expression of nearby TF *FOXA2* or other proximal genes, suggesting control of endocrine cell formation through a *trans*- rather than *cis*-regulatory mechanism. *LINC00261* was among the most highly translated lncRNAs based on ribosome profiling (Ribo-seq) and produced multiple microproteins with distinct subcellular localizations upon overexpression in vitro. To systematically assess *LINC00261*'s coding and noncoding functions, we separately introduced frameshift mutations into each of seven identified *LINC00261* sORFs. However, rigorous phenotypic characterization revealed no apparent consequences of loss of each of the seven *LINC00261*-sORF-encoded microproteins on endocrine cell development. Our comprehensive assessment of functional lncRNA translation identified a likely *trans*-regulatory role for *LINC00261* in endocrine cell differentiation that appears to be independent of the seven microproteins that were individually deleted. With this detailed investigation we provide a blueprint for the proper dissection of a gene's coding and noncoding roles in a human disease-relevant system.

## Results

### LncRNAs and nearby lineage-determining transcription factors exhibit dynamic coregulation during pancreas development

To identify lncRNAs involved in the regulation of pancreas development, we profiled RNA expression at five defined stages of hESC differentiation toward the pancreatic lineage: hESCs (ES), definitive endoderm (DE), primitive gut tube (GT), early pancreatic progenitor (PP1), and late pancreatic progenitor (PP2) (*Figure 1A*). While some lncRNAs were constitutively expressed (n = 592; 25.3%), the majority showed dynamic expression patterns (n = 1745; 74.7%), being either strongly enriched in (n = 874; 37.4%) or specific to (n = 871; 37.3%) a single developmental intermediate of pancreatic lineage progression (*Figure 1B* and *Figure 1—source data 1A*). The expression of many of these

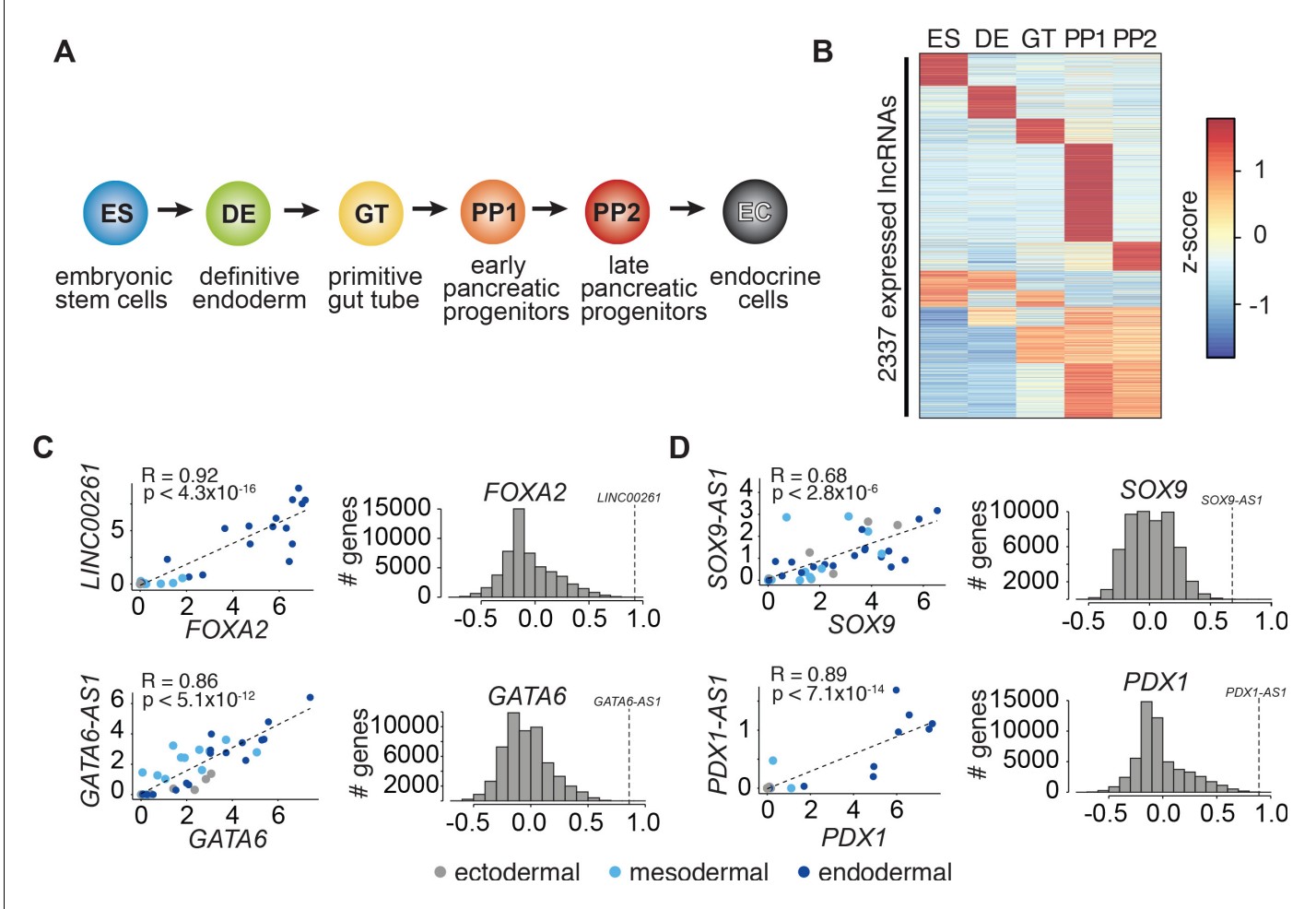

**Figure 1.** LncRNA expression and regulation during pancreatic differentiation. (**A**) Stages of directed differentiation from human embryonic stem cell (hESCs) to hormone-producing endocrine cells. The color scheme for each stage is used across all figures. (**B**) K-means clustering of all lncRNAs expressed (RPKM ≥ 1) during pancreatic differentiation based on their expression z-score (mean of n = 2 independent differentiations per stage; from CyT49 hESCs). (**C,D**) Left: Scatterplots comparing the expression of early (**C**) and late (**D**) expressed endodermal transcription factors (TFs) with the expression of their neighboring lncRNAs across 38 tissues. The dot color indicates the germ layer of origin of these tissues. Pearson correlation coefficients and p-values (t-test) are displayed. Right: Distribution of the Pearson correlation coefficients for each TF with all Ensembl 87 genes across the same 38 tissues. Dashed lines denote the correlation for the neighboring lncRNA, which for all lncRNAs shown is higher than expected by chance. See also *Figure 1—figure supplement 1* and *Figure 1—source data 1*.

The online version of this article includes the following source data and figure supplement(s) for figure 1:

**Source data 1.** Identification, regulation, and characterization of lncRNAs during pancreatic differentiation.
**Figure supplement 1.** Characterization of lncRNAs expressed during pancreatic differentiation.

dynamically regulated lncRNAs correlated with that of proximal coding genes (*Figure 1—figure supplement 1A–D* and *Figure 1—source data 1B,C*), further exemplified by a subset of lncRNAs that was specifically coregulated with the key endodermal and pancreatic TFs *GATA6*, *FOXA2*, *PDX1*, and *SOX9* (*Figure 1C,D*). The expression coregulation of these lncRNA-TF pairs is likely explained by a shared chromatin environment (*Figure 1—figure supplement 1E–H*), which raises the possibility that like the TFs, the function of the lncRNAs is also required for endoderm and pancreas development.

## Many pancreatic progenitor-expressed lncRNAs are cytoplasmically enriched and translated

Although most functional roles described for lncRNAs to date have been predominantly nuclear (*Marchese et al., 2017*), multiple recent studies have shown that many lncRNAs are cytosolic and translated into sometimes biologically active microproteins (reviewed in *Makarewich and Olson, 2017*). To further characterize the above-identified dynamically regulated lncRNAs, we analyzed their subcellular localization and translation potential using fractionation RNA-seq and Ribo-seq across multiple hESC clones independently differentiated into PP2 stage pancreatic progenitors (*Figure 2A*). Of all lncRNAs expressed in two replicate differentiations into PP2 cells, we classified 21% (n = 347) as localized to the nucleus, whereas a larger number (n = 563; 34%) primarily resided in the cytosol (*Figure 2—figure supplement 1A* and *Figure 2—source data 1A*). This subcellular distribution of pancreatic lncRNAs is in agreement with previous lncRNA localization studies by us and others (*Cabili et al., 2015*; *Clark et al., 2012*; *Sun et al., 2015*; *van Heesch et al., 2014*). LncRNAs enriched in the cytosol were expressed at higher levels than nucleus-localized lncRNAs, with expression levels similar to canonical protein-coding mRNAs (*Figure 2—figure supplement 1B*). Intriguingly, almost half (49.4%) of all cytosol-enriched lncRNAs (278 out of 563) displayed dynamic expression regulation during the differentiation of hESCs to pancreatic progenitors, raising the possibility that many lncRNAs with putative developmental functions do not act in the nucleus, but instead in the cytosol where they may be translated.

To investigate the translation potential of these cytosolic lncRNAs, we used Ribo-seq, through which we obtained exceptionally deep and high quality translatome coverage across six replicate differentiations (*Figure 2—figure supplement 1C* and *Figure 2—source data 1B*). As nearly 90% of the sequenced ribosomal footprints exhibited clear 3-nucleotide codon movement characteristic of translation (*Figure 2—figure supplement 1D–F*), these data have strong predictive value for the computational detection of non-canonical ORFs, such as upstream ORFs (uORFs) in the 5' leader sequences of mRNAs and sORFs in genes annotated as lncRNAs (*Figure 2—source data 1C*). Requiring stringent reproducibility criteria (the exact ORF needed to be detected by RiboTaper (*Calviello et al., 2016*) in at least four out of six replicates), we identified a total of 625 new sORFs in lncRNAs with a median length of 47 amino acids (aa) (*Figure 2—source data 1D*). The majority of detected sORFs (76%; n = 477/625) is currently not present in the sORFs.org database (*Olexiouk et al., 2016*). The translated sORFs are located within 285 cytosolically localized lncRNAs (25.3% of all expressed lncRNAs) (*Figure 2—figure supplement 1B*), which are expressed at higher levels than untranslated lncRNAs (*Figure 2—figure supplement 1G*) and exhibit translational efficiencies similar to mRNAs (*Figure 2—figure supplement 1H* and *Figure 2—source data 1E*). Of note, almost none of the newly identified sORFs are highly conserved across species, as judged by their low PhyloCSF scores (*Lin et al., 2011*; *Figure 2—source data 1D*).

Using approaches similar to ours, non-canonical sORFs have previously been characterized in multiple immortalized human cell lines (*Bazzini et al., 2014*; *Calviello et al., 2016*; *Chen et al., 2020*; *Ji et al., 2015*; *Martinez et al., 2020*; *Prensner et al., 2020*; *Raj et al., 2016*) and human tissues (*van Heesch et al., 2019*). However, to our knowledge, our data constitute the first comprehensive set of non-canonical human ORFs generated from a non-transformed human cell model of development, providing a valuable resource for future functional studies.

## Translated lncRNAs in pancreatic progenitors produce microproteins with distinct subcellular localizations

Having established that many stage-specific pancreatic lncRNAs are translated, we next sought to validate their translation potential through independent experimental approaches, additionally

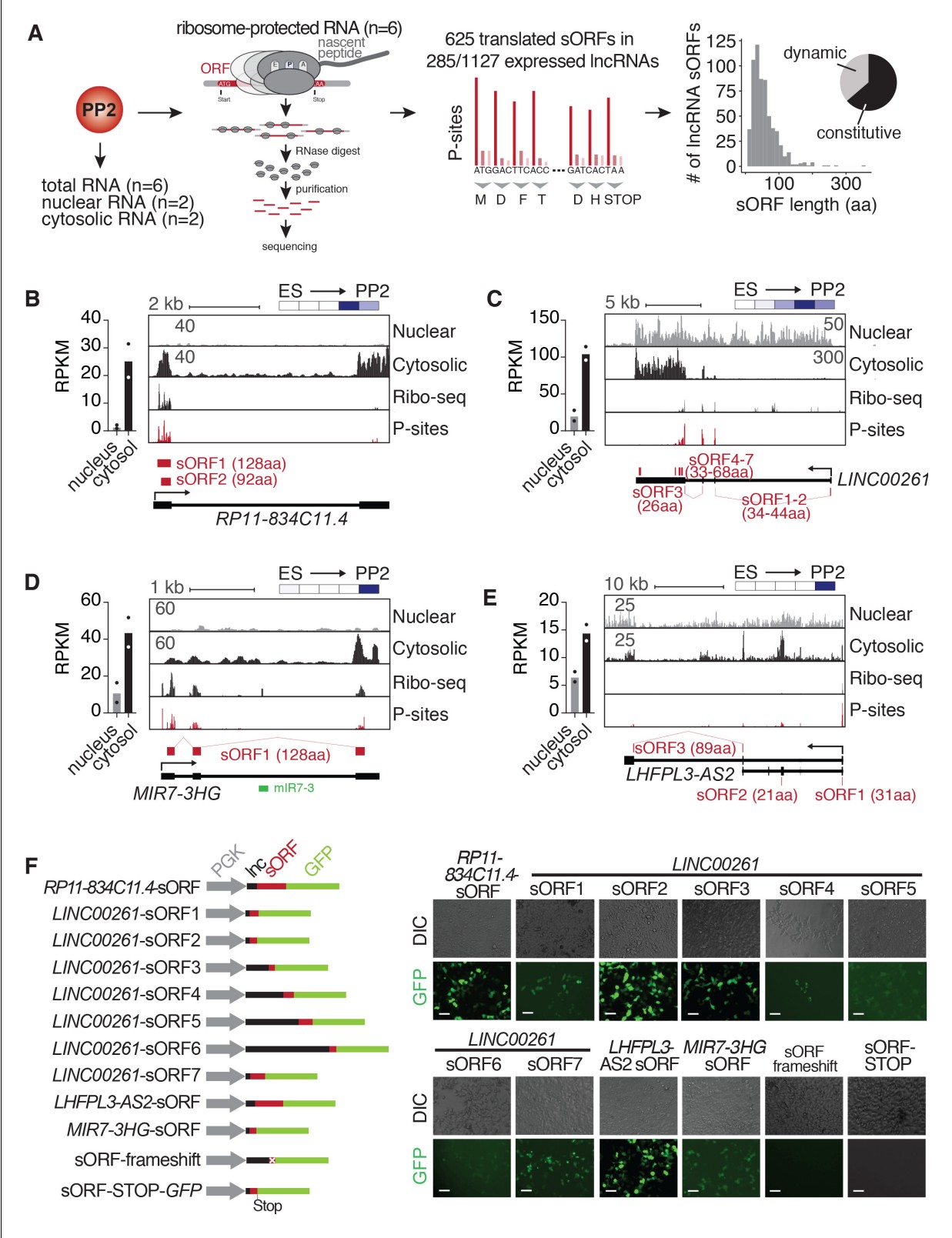

**Figure 2.** Cytosolic lncRNAs contain translated small open reading frames. (**A**) Overview of experimental strategy for subcellular fractionation and Ribo-seq-based identification of translated small open reading frames (sORFs) from lncRNAs expressed in PP2 cells. Replicates from six independent differentiations to PP2 stage each for total (polyA) RNA-seq and Ribo-seq experiments, and two biological replicates for the subcellular fractionation were analyzed. The histogram on the far right depicts the size distribution of the sORF-encoded small peptides as number of amino acids (aa). The pie

*Figure 2 continued on next page*

*Figure 2 continued*

chart summarizes the percentages of constitutively and dynamically expressed sORF-encoding lncRNAs during pancreatic differentiation of CyT49 hESCs. (B–E) Left: Bar graphs showing nuclear and cytosolic expression (in RPKM) of lncRNAs *RP11-834C11.4* (B), *LINC00261* (C), *MIR7-3HG* (D), and *LHFPL3-AS2* (E). Data are shown as mean, with individual data points represented by dots (n = 2 biological replicates). Right: Subcellular fractionation RNA-seq, Ribo-seq, and P-site tracks (ribosomal P-sites inferred from ribosome footprints on ribosome-protected RNA) for loci of the depicted lncRNAs. Identified highest stringency sORFs (ORF in 6/6 replicates) are shown in red. For *LINC00261*, visually identified sORFs 1 and 2 are also shown. Heatmaps in the top right visualize the relative expression of the shown lncRNAs during pancreatic differentiation (means of two biological replicates per stage), on a minimum (white)/maximum (dark blue) scale. (F) In vivo translation reporter assays testing whether sORFs computationally defined in (A) give rise to translation products in HEK293T cells when fused in-frame to a GFP reporter. Left: Schematic of the constructs (gray: *PGK* promoter, black: lncRNA sequence 5′ to sORF to be tested, red: sORF, green: GFP ORF). Right: Representative DIC and GFP images of HEK293T cells transiently transfected with the indicated reporter constructs. Scale bars = 50 µm. See also *Figure 2—figure supplement 1* and *Figure 2—source data 1*. The online version of this article includes the following source data and figure supplement(s) for figure 2:

**Source data 1.** RNA-seq after subcellular fractionation and Ribo-seq in PP2 cells.
**Figure supplement 1.** Cytosolic lncRNAs engage with ribosomes.

investigating the production of the predicted microproteins at the protein level. To this end, we first performed coupled in vitro transcription:translation assays on endogenous and complete transcript isoforms of four of the most highly translated lncRNAs (*LINC00261*, *RP11-834C11.4*, *LHFPL3-AS2*, and *MIR7-3HG*; *Figure 2—figure supplement 1I*; expression and ORF information in *Figure 2B–E*). Second, we generated a series of in vivo translation reporter constructs to assess the subcellular localization of microproteins translated from each of ten sORFs derived from the same four lncRNAs. Transient expression of individual constructs carrying in-frame GFP fusions in HEK293T cells produced GFP signal for all ten assayed microproteins, which was abolished upon introduction of a frameshift within the sORF or a stop codon following the sORF sequence (*Figure 2F* and *Figure 2—figure supplement 1J–L*). To rule out a possible localization bias induced by the GFP fusion, we also expressed a FLAG-tag fusion peptide (*RP11-834C11.4* sORF-1xFLAG), which revealed a cytoplasmic localization identical to the one observed for the GFP construct (*Figure 2—figure supplement 1J*). While most sORF-GFP fusion products were ubiquitously distributed throughout transfected cells, *LINC00261* sORF4-GFP specifically localized to mitochondria (*Figure 2—figure supplement 1K*), and *LINC00261* sORF7-GFP exhibited a perinuclear accumulation pattern reminiscent of aggresomes (*Figure 2—figure supplement 1L*). Taken together, our results validate the translation potential of sORFs encoded by pancreatic progenitor-expressed lncRNAs and show that, upon ectopic expression, these translation events result in the production of microproteins with different subcellular localizations.

## Deletion phenotypes of translated lncRNAs during hESC pancreatic differentiation

To identify potential functional roles of translated lncRNAs during pancreas development, we selected ten candidates for CRISPR/Cas9-based genome editing in hESCs through excision of the lncRNA promoter or entire lncRNA locus (*Figure 3A,B*). These ten lncRNAs were prioritized based on (i) high expression and endodermal tissue-specificity, (ii) dynamic regulation during pancreas development, (iii) abundant translation of sORFs, and (iv) proximity to TFs with known roles in endoderm and pancreas development. For seven of the selected lncRNAs, translation was highly abundant and reproducibly detected across Ribo-seq replicates: *LINC00617* (also known as *TUNAR*; *Lin et al., 2014*), *GATA6-AS1* (also known as *GATA6-AS*; *Neumann et al., 2018*), *LINC00261*, *RP11-834C11.4*, *SOX9-AS1*, *MIR7-3HG*, and *LHFPL3-AS2*. Although for two additional lncRNAs the translation potential could not be determined, they were nonetheless included because of a previously reported requirement for definitive endoderm formation (*DIGIT*, also known as *GSC-DT*) (*Daneshvar et al., 2016*) and genomic localization adjacent to the definitive endoderm TF *LHX1* (*RP11-445F12.1*, also known as *LHX1-DT*). Lastly, *LINC00479* was chosen as a non-translated control with expression dynamics and a subcellular localization similar to *LINC00261*. Of note, for each of the ten selected lncRNAs, we generated at least two independent hESC knockout (KO) clones and used different combinations of single guide RNAs where possible (*Figure 3—source data 1A*).

We next differentiated each of the lncRNA KO hESC lines stepwise toward the pancreatic endocrine cell stage, conducting up to 16 replicate differentiations for each KO clone. Because

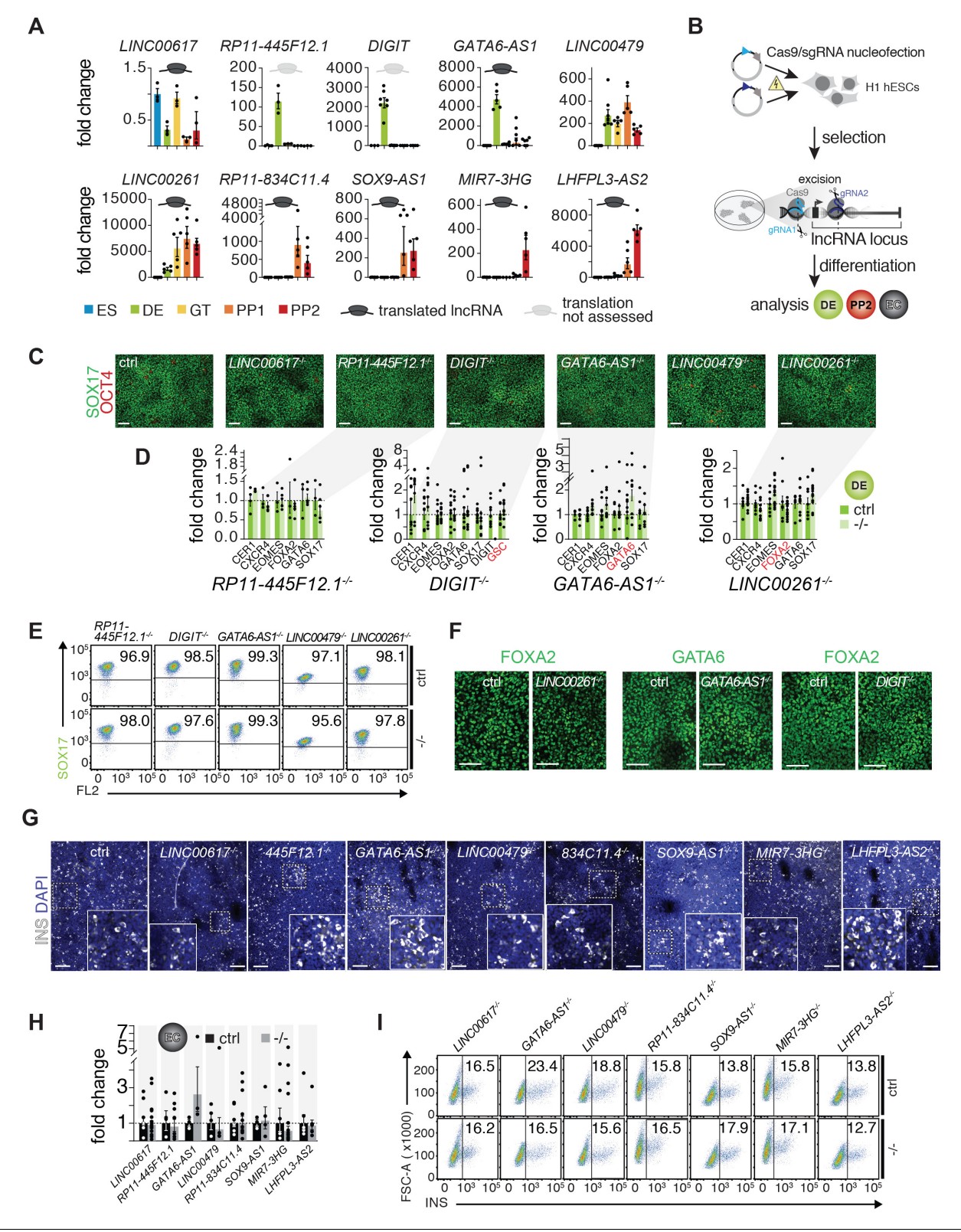

**Figure 3.** A small-scale CRISPR loss-of-function screen for dynamically expressed and translated lncRNAs during pancreatic differentiation. (**A**) qRT-PCR analysis of candidate lncRNAs during pancreatic differentiation of H1 hESCs relative to the ES stage. Data are shown as mean ± S.E.M. (mean of n = 2–6 independent differentiations per stage; from H1 hESCs). Individual data points are represented by dots. See also *Figure 3—source data 2*. (**B**) CRISPR-based lncRNA knockout (KO) strategy in H1 hESCs and subsequent phenotypic characterization. (**C**) Immunofluorescence staining for OCT4

*Figure 3 continued on next page*

*Figure 3 continued*

and SOX17 in DE from control (ctrl) and KO cells for the indicated lncRNAs (representative images, n ≥ 3 independent differentiations; at least two KO clones were analyzed). (D) qRT-PCR analysis of DE lineage markers in DE from control and lncRNA KO (-/-) cells. TF genes in cis to the lncRNA locus are highlighted in red. Data are shown as mean ± S.E.M. (n = 3–16 replicates from independent differentiations and different KO clones). Individual data points are represented by dots. NS, p-value>0.05; t-test. See also *Figure 3—source data 3*. (E) Flow cytometry analysis at DE stage for SOX17 in control and KO (-/-) cells for indicated lncRNAs. The line demarks isotype control. Percentage of cells expressing SOX17 is indicated (representative experiment, n ≥ 3 independent differentiations from at least two KO clones). (F) Immunofluorescence staining for FOXA2 or GATA6 in DE from control and *LINC00261*, *GATA6-AS1*, and *DIGIT* KO cells. (G) Immunofluorescence staining for insulin (INS) in endocrine cell stage (EC) from control and KO hESCs for the indicated lncRNAs (representative images, n ≥ 3 independent differentiations from at least two KO clones). Boxed areas (dashed boxes) are shown in higher magnification. (H) qRT-PCR analysis of *INS* in EC stage cultures from control and lncRNA KO (-/-) hESCs. Data are shown as mean ± S.E.M. (n ≥ 4 replicates from independent differentiations of at least two KO clones). Individual data points are represented by dots. NS, p-value>0.05; t-test. See also *Figure 3—source data 4* (I) Flow cytometry analysis at EC stage for INS in control and KO (-/-) cells for indicated lncRNAs. The line demarks isotype control. Percentage of cells expressing insulin is indicated (representative experiment, n ≥ 3 independent differentiations each from at least two KO clones). Scale bars = 100 μm. See also *Figure 3—figure supplement 1* and *Figure 3—source data 1–4*.

The online version of this article includes the following source data and figure supplement(s) for figure 3:

**Source data 1.** Differentially expressed genes after lncRNA deletion.
**Source data 2.** Source data used for the qRT-PCR quantification of gene expression presented in *Figure 3A*.
**Source data 3.** Source data used for the qRT-PCR quantification of gene expression presented in *Figure 3D*.
**Source data 4.** Source data used for the qRT-PCR quantification of *INS* expression presented in *Figure 3H*.
**Figure supplement 1.** Minor gene expression changes in definitive endoderm or pancreatic progenitor cells after lncRNA deletion.

*LINC00617*, *RP11-445F12.1*, *DIGIT*, *GATA6-AS1*, *LINC00479*, and *LINC00261* were first expressed at, or before, the definitive endoderm stage (*Figure 3A*), we determined whether KO hESCs for these lncRNAs exhibited defects in definitive endoderm formation. Despite efficient lncRNA depletion (*Figure 3—figure supplement 1A,B*), neither quantification of definitive endoderm marker gene expression by qRT-PCR, nor immunofluorescence staining or flow cytometric analysis of the definitive endoderm marker SOX17 showed differences indicative of impaired endoderm formation in lncRNA KO lines (*Figure 3C–E*). Importantly, expression of TFs located in the direct vicinity of these lncRNAs, including *GSC* (*DIGIT*), *LHX1* (*RP11-445F12.1*), *GATA6* (*GATA6-AS1*), and *FOXA2* (*LINC00261*), was unaffected by the lncRNA KO (*Figure 3F*, *Figure 3—figure supplement 1C*, *Figure 3—source data 1B–D*), arguing against *cis*-regulation by these lncRNAs. These findings are in contrast to prior reports that have shown a requirement for *LINC00261* and *DIGIT* in definitive endoderm formation and the regulation of neighboring TFs *FOXA2* and *GSC*, respectively (*Amaral et al., 2018*; *Daneshvar et al., 2016*; *Jiang et al., 2015*; *Swarr et al., 2019*).

Next, we further differentiated control and KO lines for eight out of ten lncRNAs toward the endocrine cell stage, excluding *DIGIT* and *RP11-445F12.1* because they are not expressed after the definitive endoderm stage (*Figure 3A*). In KO hESC lines of seven out of these eight lncRNAs, we observed no effect on pancreatic progenitor cell formation or gene expression, with the exception of a handful of dysregulated genes in *LHFPL3-AS2* and *RP11-834C11.4* KO cells (*Figure 3—figure supplement 1C* and *Figure 3—source data 1E–K*). Furthermore, deletion of seven out of the eight lncRNAs did not impair endocrine cell formation, as determined by quantification of insulin[+] cells and insulin mRNA levels (*Figure 3G–I*). Similar to the RNA expression results obtained at the definitive endoderm stage, deletion of none of the lncRNAs close to pancreatic TFs (e.g. *GATA6-AS1* and *SOX9-AS1*) altered the expression of these TFs, once more arguing against *cis*-regulation of these TFs by the neighboring lncRNA (*Figure 3—figure supplement 1C*). Thus, nine out of ten endoderm- and pancreatic progenitor-enriched lncRNAs functionally investigated here appear to be nonessential for induction of the pancreatic fate and formation of insulin[+] cells. Furthermore, these lncRNAs do not appear to control the transcript levels of proximal TFs.

## *LINC00261* knockout impairs endocrine cell development

The exception was the endoderm-specific lncRNA *LINC00261*, which is highly expressed and translated in pancreatic progenitors (*Figure 4—figure supplement 1A* and *Figure 2C*). While deletion of *LINC00261* caused no discernable phenotype in definitive endoderm (*Figure 3C–F* and *Figure 3—figure supplement 1C*), we observed a significant 30–50% reduction in the number of insulin[+] cells at the endocrine cell stage (*Figure 4A,B*). This reduction in insulin[+] cell numbers was consistent

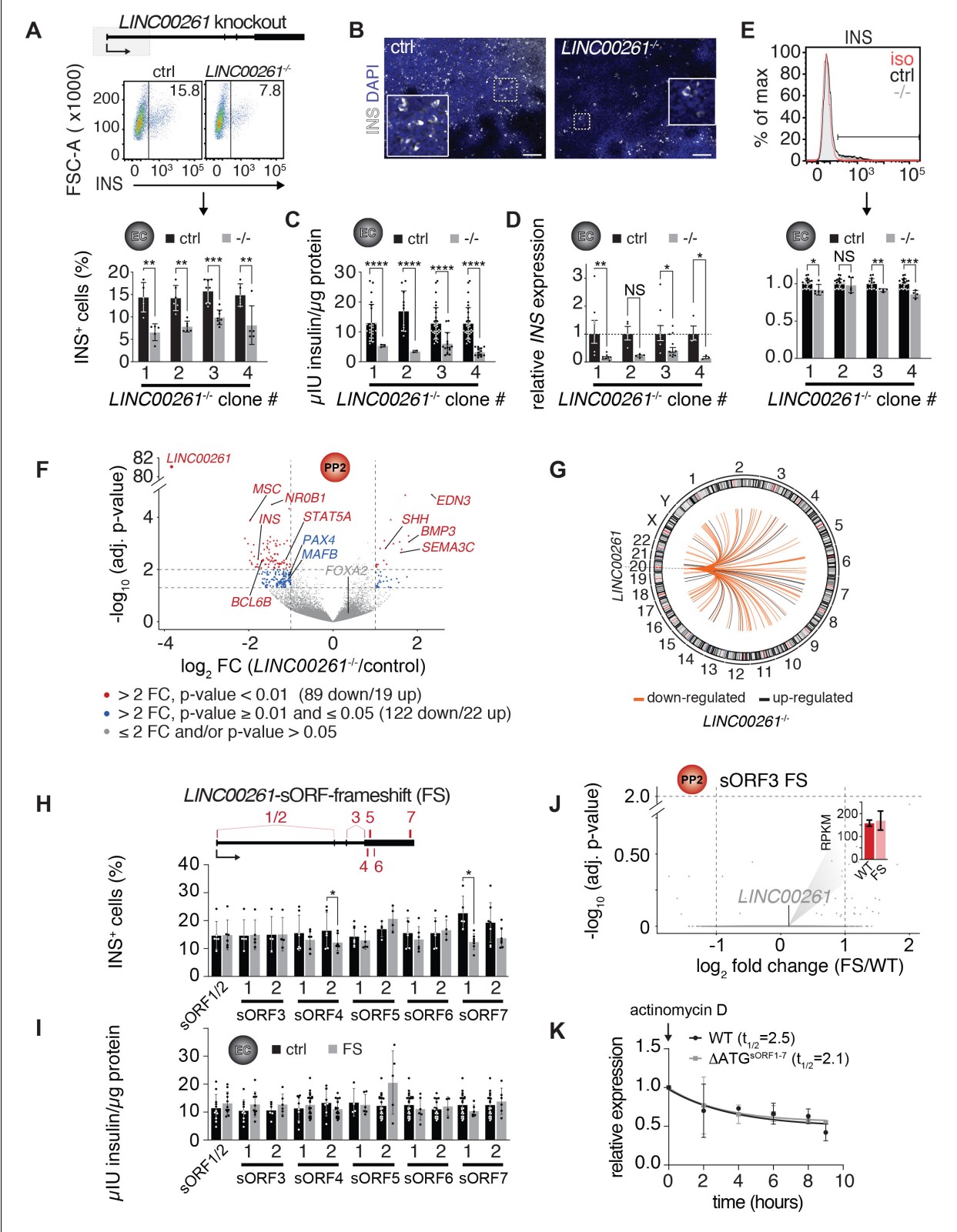

**Figure 4.** *LINC00261* deletion impedes pancreatic endocrine cell differentiation. (**A**) Flow cytometry analysis at endocrine cell stage (EC) for insulin (INS) in control (ctrl) and *LINC000261⁻/⁻* H1 hESCs. Top panel: Schematic of the *LINC00261* locus. The dashed box represents the genomic deletion. Middle panel: The line demarks isotype control. Percentage of cells expressing INS is indicated (representative experiment, n = 4 deletion clones generated with independent sgRNAs). Bottom panel: Bar graph showing percentages of INS-positive cells. Data are shown as mean ± S.D. (n = 5 (clone

*Figure 4 continued on next page*

Figure 4 continued

1), n = 6 (clone 2), n = 8 (clone 3), n = 5 (clone 4) independent differentiations). Individual data points are represented by dots. (B) Immunofluorescence staining for INS in EC stage cultures from control and *LINC000261*[-/-] hESCs (representative images, number of differentiations see A). Boxed areas (dashed boxes) are shown in higher magnification. (C) ELISA for INS in EC stage cultures from control and *LINC00261*[-/-] hESCs. Data are shown as mean ± S.D. (n = 3 (clone 1), n = 2 (clone 2), n = 14 (clone 3), n = 13 (clone 4) independent differentiations). Individual data points are represented by dots. (D) qRT-PCR analysis of *INS* in EC stage cultures from control and *LINC00261*[-/-] hESCs. Data are shown as mean ± S.E.M. (n = 8 (clone 1), n = 4 (clone 2), n = 10 (clone 3), n = 3 (clone 4) independent differentiations). Individual data points are represented by dots. (E) Quantification of median fluorescence intensity after INS staining of control and *LINC00261*[-/-] EC stage cultures. Data are shown as mean ± S.D. (n = 5 (clone 1), n = 5 (clone 2), n = 4 (clone 3), n = 4 (clone 4) independent differentiations). iso, isotype control. Individual data points are represented by dots. (F) Volcano plot displaying gene expression changes in control versus *LINC00261*[-/-] PP2 cells (n = 6 independent differentiations from all four deletion clones). Differentially expressed genes are shown in red (DESeq2;>2 fold change (FC), adjusted p-value<0.01) and blue (>2 fold change, adjusted p-value≥0.01 and≤0.05). Thresholds are represented by vertical and horizontal dashed lines. *FOXA2* in cis to *LINC00261* is shown in gray (gray dots represent genes with ≤ 2 fold change and/or adjusted p-value>0.05). (G) Circos plot visualizing the chromosomal locations of the 108 genes differentially expressed (DESeq2;>2 fold change (FC), adjusted p-value<0.01) in *LINC00261*[-/-] compared to control PP2 cells, relative to *LINC00261* on chromosome 20. No chromosome was over- or underrepresented (Fisher test, p-value>0.05 for all chromosomes). (H) Top panel: Schematic of the *LINC00261* locus, with the location of its sORFs (1 to 7) marked by vertical red bars. Bottom panel: Flow cytometric quantification of INS-positive cells in control and *LINC00261*-sORF-frameshift (FS) at the EC stage. Data are shown as mean ± S.D. (n = 4–7 independent differentiations per clone). (I) ELISA for INS in EC stage cultures from control and *LINC00261*-sORF-FS hESCs. Data are shown as mean ± S.D. (n = 3–7 independent differentiations per clone). (J) Volcano plot displaying gene expression changes in control versus *LINC00261*-sORF3-FS PP2 cells. No gene was differentially expressed (DESeq2;>2 fold change, adjusted p-value<0.01; indicated by dashed horizontal and vertical lines; n = 2 independent differentiations). *LINC00261* is shown in gray, the bar graph insert displays *LINC00261* RPKM values in control and *LINC00261*-sORF3-FS PP2 cells. (K) *LINC00261* half-life measurements in HEK293T cells transduced with lentivirus expressing either wild type (WT) *LINC00261* or ΔATG[sORF1-7] *LINC00261* (mutant in which the ATG start codons of sORFs 1–7 were changed to non-start codons). HEK293T were treated with the transcription inhibitor actinomycin D and RNA isolated at 0, 2, 4, 6, 8, and 9 hr post actinomycin D addition. *LINC00261* expression was analyzed by qRT-PCR relative to the *TBP* gene. Data are shown as mean ± S.E.M. (n = 3 biological replicates for each assay time point). *, p-value<0.05; **, p-value<0.01; ***, p-value<0.001; ****, p-value<0.0001; NS, p-value>0.05; t-test. Scale bars = 100 μm. See also *Figure 4—figure supplement 1* and *Figure 4—source data 1–3*.

The online version of this article includes the following source data and figure supplement(s) for figure 4:

**Source data 1.** Characterization of *LINC00261* knockout and *LINC00261*-sORF3-frameshift PP2 cells.
**Source data 2.** List of oligonucleotides and synthetic gene fragments used in this study.
**Source data 3.** Source data used for the insulin measurements presented in *Figure 4*.
**Figure supplement 1.** Characterization of *LINC00261*-deleted pancreatic progenitor cells.

across four separately derived *LINC00261* KO hESC lines, each independently differentiated to endocrine cell stage 5–8 times. In agreement with reduced insulin[+] cell numbers, insulin content and insulin mRNA levels were also reduced in *LINC00261* KO endocrine stage cultures (*Figure 4C,D*). Analysis of insulin median fluorescence intensities by flow cytometry further showed no reduction in insulin levels per cell in one *LINC00261* KO clone and a mild reduction in the three other clones (*Figure 4E*), indicating that *LINC00261* predominately regulates endocrine cell differentiation rather than maintenance of insulin production in beta cells.

To determine the molecular effects of *LINC00261* deletion, we performed RNA-seq in pancreatic progenitors derived from *LINC00261* KO and control hESCs. Similar to the absence of *cis*-regulatory functions observed in the other lncRNA KOs, we found no evidence for *cis*-regulation of *FOXA2* by *LINC00261* (*Figure 4F* and *Figure 4—figure supplement 1C*). However, we observed downregulation of the TFs *MAFB* and *PAX4* (*Figure 4F*, *Figure 4—figure supplement 1B*, *Figure 4—source data 1A*), which are important regulators of beta cell differentiation (*Artner et al., 2007*; *Sosa-Pineda et al., 1997*). Of note, genes differentially expressed in *LINC00261* KO cells mapped to all chromosomes and showed no enrichment for chromosome 20 where *LINC00261* resides (*Figure 4G*). These results suggest a *trans*- rather than *cis*-regulatory function for *LINC00261*, consistent with its predominantly cytosolic localization, translation, and diffuse distribution within the nucleus (*Figure 2C* and *Figure 4—figure supplement 1D*). *Trans*-regulatory roles of *LINC00261* have also been observed in previous studies (*Aguet et al., 2019*; *Shi et al., 2019*; *Wang et al., 2019*; *Wang et al., 2017*; *Yan et al., 2019*). This potential *trans* functionality prompted us to further investigate whether *LINC00261*'s coding or noncoding features are essential for endocrine cell differentiation.

## One-by-one disruption of *LINC00261*'s sORFs does not impact endocrine cell differentiation

We established that *LINC00261* harbors multiple distinct and highly translated sORFs, which raises the possibility that the translation of these sORFs is functionally important for endocrine cell differentiation. To systematically discriminate *LINC00261*'s coding and noncoding roles, we individually mutated its seven most highly translated sORFs independently in hESCs, leaving the lncRNA sequence, and hence any noncoding function coupled to RNA sequence or structure, grossly intact. Each of these hESC lines either carried a homozygous frameshift mutation near the microprotein's N-terminus (for sORFs 1–6) or a full sORF deletion (sORF7; *Figure 4—source data 1B*). After verifying that CRISPR editing of the *LINC00261* locus did not impact *LINC00261* transcript levels (*Figure 4—figure supplement 1E*), we quantified (i) insulin mRNA levels, (ii) insulin⁺ cells, and (iii) total insulin content in endocrine cell stage cultures. We observed no difference between sORF loss-of-function and control hESC lines for most of these endpoints (*Figure 4H,I* and *Figure 4—figure supplement 1E*), although we noticed that the number of insulin⁺ cells, but not the amount of insulin produced, was reduced in one of the two sORF4 and sORF7 KO clones. Transcriptome analysis of pancreatic progenitors with frameshifts in sORF3 (the most highly translated *LINC00261*-sORF; *Figure 2C* and *Figure 2—source data 1D*) revealed no differentially expressed genes between *LINC00261*-sORF3 frameshift and control cells (*Figure 4J* and *Figure 4—source data 1C*), contrasting observations in *LINC00261* RNA KO pancreatic progenitors (*Figure 4F* and *Figure 4—source data 1A*). These results indicate that there is not one dominant *LINC00261* sORF that is required for endocrine cell formation, suggesting a functional role of the *LINC00261* transcript and not the individual sORFs mutated here. However, it is possible that the different sORFs, or the microproteins translated from these sORFs, are functionally redundant and capable of phenotypic rescue.

It has been suggested that ribosome association can control lncRNA transcript levels by inducing nonsense-mediated decay (NMD) (*Carlevaro-Fita et al., 2016*; *Tani et al., 2013*). Therefore, we determined whether the presence of multiple sORFs could regulate *LINC00261* stability. To this end, we simultaneously mutated start codons of all seven sORFs (ΔATG$^{sORF1-7}$ *LINC00261*) and expressed either wild type or ΔATG$^{sORF1-7}$ *LINC00261* ectopically in HEK293T cells, where *LINC00261* is normally not expressed. *LINC00261* half-life measurements upon transcriptional inhibition with actinomycin D revealed no difference in *LINC00261* levels between wild type and ΔATG$^{sORF1-7}$ *LINC00261* (*Figure 4K*), suggesting that the translation of the seven sORFs does not reduce *LINC00261* transcript stability.

In sum, through the systematic, one-by-one removal of sORFs within a highly translated lncRNA with functional importance for pancreatic endocrine cell formation, we found no evidence to implicate the individual sORFs, or the microproteins they produce, in endocrine cell development. Although *LINC00261*'s sORFs may share functional redundancy or have developmental roles that do not affect the production of insulin⁺ cells, our findings strongly suggest that by themselves, these sORFs are not functionally required for endocrine cell formation.

## Discussion

### Limited *cis*-regulatory consequences of lncRNA deletion

In this study we globally characterized molecular features of lncRNAs expressed during progression of hESCs toward the pancreatic lineage, including their subcellular localization and potential to be translated. We performed a phenotypic CRISPR loss-of-function screen, focusing on ten developmentally regulated, highly expressed, and highly translated lncRNAs proximal to TFs known to regulate pancreas development. The first important observation from this screen is that we found no evidence to implicate the lncRNAs *LINC00261*, *DIGIT*, *GATA6-AS1*, *SOX9-AS1*, and *RP11-445F12.1* in the *cis*-regulation of their neighboring TFs *FOXA2*, *GSC*, *GATA6*, *SOX9*, and *LHX*, respectively, despite tight transcriptional coregulation of the lncRNA-TF pairs.

Contrasting our findings, a number of studies have reported *cis*-regulation of *FOXA2* by *LINC00261* (*Amaral et al., 2018*; *Jiang et al., 2015*; *Swarr et al., 2019*). However, several lines of evidence strongly support the conclusion that *FOXA2* is not regulated by *LINC00261* in our experimental system. First, we examined *FOXA2* mRNA expression in *LINC000261*$^{-/-}$ cells at both the definitive endoderm and pancreatic progenitor cell stages. Second, we analyzed *FOXA2* expression

using two independent methods, namely qRT-PCR and RNA-seq. Third, immunofluorescence staining in definitive endoderm revealed no difference in FOXA2 protein expression between control and LINC00261[-/-] cells.

While different cellular contexts and species could explain the discrepancy between our findings and the ones by *Amaral et al., 2018* and *Swarr et al., 2019*, *Jiang et al., 2015* reported *FOXA2* regulation by *LINC00261* in hESC-derived definitive endoderm. One important difference between our study and the study by Jiang et al. is that we employed CRISPR-Cas9-mediated deletion, whereas Jiang et al. used shRNA-mediated knockdown to inactivate *LINC00261*. It is possible that lncRNA deletion triggers compensatory mechanisms that are not activated after shRNA-mediated knockdown. For coding genes, mutant mRNA degradation has been shown to trigger genetic compensation (*El-Brolosy et al., 2019*). Another difference between our study and the one by Jiang et al. is that our differentiation protocol was more efficient in generating definitive endoderm. It is conceivable that the stability of the cell fate and identity of neighboring cells could influence how *LINC00261* loss-of-function affects gene regulation.

## Translation of short, non-canonical ORFs in lncRNAs: regulatory, microprotein-producing, or just tolerated?

Although lncRNAs are now appreciated as a novel and abundant source of sORF-encoded biologically active microproteins (*Makarewich and Olson, 2017*), it remains largely unknown which translation events lead to the production of microproteins, which solely have regulatory potential, or which have no functional roles, but are not negatively selected against. The cytosolic localization and translation of many RNAs classified as lncRNAs provides a strong rationale for considering both, coding and noncoding functions.

In this study, we identified the translated lncRNA *LINC00261* as a novel regulator of pancreatic endocrine cell differentiation, as evidenced by a severe reduction in insulin[+] cell numbers upon *LINC00261* deletion. We show that *LINC00261* transcripts are highly abundant in pancreatic progenitors and, albeit present in the nucleus, are predominantly localized to the cytoplasm. Here, they frequently associate with ribosomes which leads to the translation of multiple independent sORFs. We show that the sORFs are capable of producing microproteins with distinct subcellular localizations upon expression in vitro. In contrast to *LINC00261* deletion, individual frameshift mutations in each of *LINC00261*'s sORFs did not impair endocrine cell development, suggesting that the requirement of *LINC00261* for endocrine cell development can be uncoupled from the translation of its multiple sORFs. However, this does not exclude the possibility that these sORFs or the microproteins they produce could possess functions that become relevant under specific environmental, developmental, or disease conditions not examined in this study.

We found that mutating all translated *LINC00261* sORFs simultaneously, thereby likely reducing *LINC00261*'s ability to bind ribosomes, did not affect *LINC00261* transcript levels in HEK293T cells. This indicates that, in contrast to reports suggesting that translated sORFs can regulate RNA stability by promoting nonsense-mediated RNA decay (*Carlevaro-Fita et al., 2016*; *Tani et al., 2013*), the high translation levels and multiple sORFs of *LINC00261* are unlikely to be part of a *LINC00261* decay pathway. It would have been interesting to determine how concurrent mutation of all sORFs in *LINC00261* affects pancreatic cell differentiation. However, given the size of the *LINC00261* locus and the many sORFs, such an approach comes with technical challenges and significant caveats.

## *LINC00261* - a potential *trans* regulator of endocrine cell differentiation?

Several lines of evidence suggest that *LINC00261* regulates endocrine cell differentiation in trans: (i) *LINC00261* transcripts show a diffuse distribution in multiple subcellular compartments, (ii) genes differentially expressed in *LINC00261* KO cells are randomly distributed throughout the genome, (iii) expression of the nearby TF *FOXA2* is not affected by *LINC00261* deletion. Such a *trans* regulatory mechanism for *LINC00261* is supported by a recent study from the GTEx Consortium, where *LINC00261* is highlighted as one of a few lncRNAs that forms a potential *trans* regulatory hotspot through genetic interactions that influence the expression of multiple distant genes (*Aguet et al., 2019*). Consistent with its preferential cytosolic localization, and further supporting the notion of a *trans* regulatory mechanism, *LINC00261* has been suggested to regulate gene expression through

non-nuclear mechanisms, e.g. by preventing nuclear translocation of β-catenin (*Wang et al., 2017*) or by acting as a miRNA sponge (*Shi et al., 2019*; *Wang et al., 2019*; *Yan et al., 2019*). Although our observations and current literature strongly hint to a function in trans independent of the produced microproteins, the exact mechanism by which *LINC00261* regulates gene expression in pancreatic progenitors remains to be determined.

### Limitations and future directions

In this study, we have characterized the role of translated lncRNAs, and in particular *LINC00261,* in a hESC differentiation system that mimics pancreas development. However, there are several potential limitations that need to be considered when interpreting the results. First, a small subset of analyses in this study was based on low numbers of replicate differentiations, in particular the cytosolic versus nuclear fractionation RNA-seq experiments, where only two replicate differentiations into pancreatic progenitor cells were analyzed. Second, although we provide evidence that *LINC00261* can produce microproteins using Ribo-seq, which is further supported by in vitro translation assays and overexpression of *LINC00261* constructs with different in-frame tags, we provide no protein-level evidence for the endogenous production and stability of *LINC00261*'s microproteins in this differentiation system or in human pancreas development in vivo. Moreover, due to its highly specific expression pattern, *LINC00261* has not been previously detected by sORF analyses in other cell types (*Bazzini et al., 2014*; *Calviello et al., 2016*; *Chen et al., 2020*; *Ji et al., 2015*; *Martinez et al., 2020*; *Prensner et al., 2020*; *Raj et al., 2016*; *van Heesch et al., 2019*). Even though we show microprotein production in vitro, it is possible that the act of translation has a key regulatory role rather than the protein products of *LINC00261*'s sORFs. Lastly, *LINC00261*'s microproteins and sORFs may have redundant functions, which could explain why deletion of individual sORFs produces no apparent phenotype. Thus, despite limited sequence similarity and stark differences in translation rates between the identified translated sORFs in *LINC00261*, we cannot rule out that different microproteins produced by *LINC00261* compensate when one sORF is deleted. Future studies of *LINC00261*'s precise mechanisms of action could be aimed at further dissecting the potential regulatory features of sORF translation and possibility of redundancy between sORFs.

### Conclusions

In summary, we here present a rigorous, in-depth characterization of dynamically regulated and translated lncRNAs in a disease-relevant cell model of human developmental progression. Our combination of ultra-high-coverage RNA- and Ribo-seq, in vitro protein-level validation of microprotein production and localization, and the systematic, one-by-one deletion of all individual microproteins encoded by a single translated lncRNA, not only provides a detailed resource of translated 'non-canonical' sORFs and their microproteins in pancreatic development, but also serves as a blueprint for the systematic functional interrogation of translated lncRNAs.

## Materials and methods

**Key resources table**

| Reagent type (species) or resource | Designation | Source or reference | Identifiers | Additional information |
|---|---|---|---|---|
| Gene (*Homo sapiens*) | *LINC00617; TUNAR* | Ensembl 87 | ENSG00000250366 | |
| Gene (*Homo sapiens*) | *RP11-445F12.1; LHX-DT* | Ensembl 87 | ENSG00000250366 | |
| Gene (*Homo sapiens*) | *DIGIT; GSC-DT* | HGNC and NCBI RefSeq | HGNC:53074; NCBI RefSeq 108868751 | |
| Gene (*Homo sapiens*) | *GATA6-AS1; GATA6-AS* | Ensembl 87 | ENSG00000277268 | |
| Gene (*Homo sapiens*) | *LINC00479* | Ensembl 87 | ENSG00000236384 | |

*Continued on next page*

*Continued*

| Reagent type (species) or resource | Designation | Source or reference | Identifiers | Additional information |
|---|---|---|---|---|
| Gene (*Homo sapiens*) | *LINC00261; DEANR1; ALIEN; onco-lncRNA-17; lnc-FOXA2-2* | Ensembl 87 | ENSG00000236384 | |
| Gene (*Homo sapiens*) | *RP11-834C11.4* | Ensembl 87 | ENSG00000250742 | |
| Gene (*Homo sapiens*) | *SOX9-AS1* | Ensembl 87 | ENSG00000234899 | |
| Gene (*Homo sapiens*) | *MIR7-3HG* | Ensembl 87 | ENSG00000176840 | |
| Gene (*Homo sapiens*) | *LHFPL3-AS2* | Ensembl 87 | ENSG00000225329 | |
| Strain, strain background (*Escherichia coli*) | Stbl3 | ThermoFisher Scientific | Cat# C737303 | Chemically competent cells |
| Strain, strain background (*Escherichia coli*) | DH5α | New England Biolabs | Cat# C2987I | Chemically competent cells |
| Cell line (*Homo sapiens*) | H1 (embryonic stem cells) | WiCell Research Institute | NIHhESC-10–0043, RRID:CVCL_9771 | |
| Cell line (*Homo sapiens*) | HEK293T (embryonic kidney) | ATCC | Cat# CRL-3216, RRID:CVCL_0063 | |
| Antibody | anti-human OCT-4A (Rabbit monoclonal) | Cell Signaling Technology | Cat# 2890, RRID:AB_2167725 | IF (1:1000) |
| Antibody | anti-human SOX17 (Goat polyclonal) | R and D Systems | Cat# AF1924, RRID:AB_355060 | IF (1:250) |
| Antibody | anti-human FOXA2 (Goat polyclonal) | Santa Cruz Biotechnology | Cat# sc-6554, RRID:AB_2262810 | IF (1:250) |
| Antibody | anti-human GATA6 (Goat polyclonal) | Santa Cruz Biotechnology | Cat# sc-9055, RRID:AB_2108768 | IF (1:50) |
| Antibody | anti-human Insulin (Guinea pig polyclonal) | Dako | Cat# A0564, RRID:AB_10013624 | IF (1:1000) |
| Antibody | Alexa Fluor 488 AffiniPure anti-Goat IgG (Donkey polyclonal) | Jackson ImmunoResearch Labs | Cat# 706-545-148, RRID:AB_2340472 | IF (1:1000) |
| Antibody | Alexa Fluor 488 AffiniPure anti-Rabbit IgG (Donkey polyclonal) | Jackson ImmunoResearch Labs | Cat# 711-545-152, RRID:AB_2313584 | IF (1:1000) |
| Antibody | Cy3 AffiniPure anti-Goat IgG (Donkey polyclonal) | Jackson ImmunoResearch Labs | Cat# 705-165-147, RRID:AB_2307351 | IF (1:1000) |

*Continued*

| Reagent type (species) or resource | Designation | Source or reference | Identifiers | Additional information |
|---|---|---|---|---|
| Antibody | anti-human Insulin-PE (Rabbit monoclonal) | Cell Signaling Technology | Cat# 8508S, RRID:AB_11179076 | Flow cytometry (1:50) |
| Antibody | anti-human SOX17-PE (Mouse monoclonal) | BD Biosciences | Cat# 561591, RRID:AB_10717121 | Flow cytometry (5 ul per test) |
| Antibody | IgG-PE (Rabbit monoclonal) | Cell Signaling Technology | Cat# 5742S, RRID:AB_10694219 | Flow cytometry isotype control antibody (1:50) |
| Antibody | IgG1, κ antibody (Mouse monoclonal) | BD Biosciences | Cat# 556650, RRID:AB_396514 | Flow cytometry isotype control antibody (1:50) |
| Recombinant DNA reagent | pSpCas9(BB)—2A-Puro (Px459; V2.0) | Feng Zhang | RRID:Addgene_62988 | Cas9 from *S. pyogenes* with 2A-Puro, and cloning backbone for sgRNA |
| Recombinant DNA reagent | pSpCas9(BB)—2A-GFP (PX458) | Feng Zhang | RRID:Addgene_48138 | Cas9 from *S. pyogenes* with 2A-EGFP, and cloning backbone for sgRNA |
| Recombinant DNA reagent | pCMVR8.74 | Didier Trono | RRID:Addgene_22036 | 2nd generation lentiviral packaging plasmid |
| Recombinant DNA reagent | pMD2.G | Didier Trono | RRID:Addgene_12259 | VSV-G envelope expressing plasmid |
| Recombinant DNA reagent | pENTR/D-TOPO-*LINC00261* | *Kurian et al., 2015* (PMID:25739401) | | Leo Kurian (University of Cologne) |
| Recombinant DNA reagent | pRRLSIN.cPPT. PGK-GFP.WPRE | Didier Trono | RRID:Addgene_12252 | 3rd generation lentiviral backbone |
| Recombinant DNA reagent | pRRLSIN.cPPT.PGK-*LINC00261*.WPRE | This paper | | Transient (transfection) or stable (lentiviral integration) expression of wild type *LINC00261* |
| Recombinant DNA reagent | pRRLSIN.cPPT. PGK-ΔATG$^{sORF1-7}$ *LINC00261*.WPRE | This paper | | Transient (transfection) or stable (lentiviral integration) expression of ΔATG$^{sORF1-7}$ *LINC00261* |
| Recombinant DNA reagent | pRRLSIN.cPPT.PGK-*LINC00261-sORF1-GFP*.WPRE | This paper | | Transient (transfection) or stable (lentiviral integration) expression of *LINC00261-sORF1-GFP* fusion protein |
| Recombinant DNA reagent | pRRLSIN.cPPT.PGK-*LINC00261-sORF2-GFP*.WPRE | This paper | | Transient (transfection) or stable (lentiviral integration) expression of *LINC00261-sORF2-GFP* fusion protein |
| Recombinant DNA reagent | pRRLSIN.cPPT. PGK-*LINC00261-sORF3-GFP*.WPRE | This paper | | Transient (transfection) or stable (lentiviral integration) expression of *LINC00261-sORF3-GFP* fusion protein |

*Continued on next page*

*Continued*

| Reagent type (species) or resource | Designation | Source or reference | Identifiers | Additional information |
|---|---|---|---|---|
| Recombinant DNA reagent | pRRLSIN.cPPT. PGK-*LINC00261-sORF4-GFP*.WPRE | This paper | | Transient (transfection) or stable (lentiviral integration) expression of *LINC00261-sORF4-GFP* fusion protein |
| Recombinant DNA reagent | pRRLSIN.cPPT. PGK-*LINC00261-sORF5-GFP*.WPRE | This paper | | Transient (transfection) or stable (lentiviral integration) expression of *LINC00261-sORF5-GFP* fusion protein |
| Recombinant DNA reagent | pRRLSIN.cPPT. PGK-*LINC00261-sORF6-GFP*.WPRE | This paper | | Transient (transfection) or stable (lentiviral integration) expression of *LINC00261-sORF6-GFP* fusion protein |
| Recombinant DNA reagent | pRRLSIN.cPPT. PGK-*LINC00261-sORF7-GFP*.WPRE | This paper | | Transient (transfection) or stable (lentiviral integration) expression of *LINC00261-sORF7-GFP* fusion protein |
| Recombinant DNA reagent | pRRLSIN.cPPT. PGK-*LINC00261-sORF3-FS-GFP*.WPRE | This paper | | *LINC00261-sORF3-frameshift-GFP* control plasmid |
| Recombinant DNA reagent | pRRLSIN.cPPT. PGK-*LINC00261-sORF2-STOP-GFP*.WPRE | This paper | | *LINC00261-sORF2-STOP-GFP* control plasmid |
| Recombinant DNA reagent | pRRLSIN.cPPT. PGK-*RP11-834C11.4-sORF-GFP*.WPRE | This paper | | Transient (transfection) or stable (lentiviral integration) expression of *RP11-834C11.4-sORF-GFP* fusion protein |
| Recombinant DNA reagent | pRRLSIN.cPPT. PGK-*RP11-834C11.4-sORF-FLAG*.WPRE | This paper | | Transient (transfection) or stable (lentiviral integration) expression of *RP11-834C11.4-sORF-FLAG* fusion protein |
| Recombinant DNA reagent | pRRLSIN.cPPT. PGK-*MIR7-3HG-sORF-GFP*.WPRE | This paper | | Transient (transfection) or stable (lentiviral integration) expression of *MIR7-3HG-sORF-GFP* fusion protein |
| Recombinant DNA reagent | pRRLSIN.cPPT.PGK-*RP11-LHFPL3-AS2-sORF-GFP*.WPRE | This paper | | Transient (transfection) or stable (lentiviral integration) expression of *LHFPL3-AS2-sORF-GFP* fusion protein |
| Peptide, recombinant protein | Activin A | R and D Systems | Cat# 338-AC/CF | |
| Peptide, recombinant protein | Wnt3a | R and D Systems | Cat# 1324-WN-010 | |
| Peptide, recombinant protein | KGF/FGF7 | R and D Systems | Cat# 251 KG | |

*Continued on next page*

*Continued*

| Reagent type (species) or resource | Designation | Source or reference | Identifiers | Additional information |
|---|---|---|---|---|
| Peptide, recombinant protein | Noggin | R and D Systems | Cat# 3344 NG-050 | |
| Commercial assay or kit | RNeasy Mini Kit | QIAGEN | Cat# 15596018 | |
| Commercial assay or kit | RNA Clean and Concentrator−25 | Zymo Research | Cat# R1018 | |
| Commercial assay or kit | Paris Kit | Thermo Fisher Scientific | Cat# AM1921 | |
| Commercial assay or kit | RNase-Free DNase Set (50) | QIAGEN | Cat# 79254 | |
| Commercial assay or kit | TURBO DNA-free Kit | Thermo Fisher Scientific | Cat# AM1907 | |
| Commercial assay or kit | TruSeq Ribo Profile (Mammalian) Library Prep Kit | Illumina | Cat# RPYSC12116 | |
| Commercial assay or kit | TruSeq Stranded mRNA Library Prep | Illumina | Cat# 20020594 | |
| Commercial assay or kit | TruSeq Stranded Total RNA Library Prep Gold | Illumina | Cat# 20020599 | |
| Commercial assay or kit | KAPA mRNA HyperPrep Kit | Roche | Cat# KK8581 | |
| Commercial assay or kit | High Sensitivity D1000 ScreenTape | Agilent Technologies | Cat# 5067–5584 | |
| Commercial assay or kit | RNA ScreenTape | Agilent Technologies | Cat# 5067–5576 | |
| Commercial assay or kit | RNA ScreenTape Sample Buffer | Agilent Technologies | Cat# 5067–5577 | |
| Commercial assay or kit | RNA ScreenTape Ladder | Agilent Technologies | Cat# 5067–5578 | |
| Commercial assay or kit | Qubit ssDNA assay kit | Thermo Fisher Scientific | Cat# Q10212 | |
| Commercial assay or kit | KOD Xtreme DNA Hotstart Polymerase | Millipore | Cat# 71975 | |
| Commercial assay or kit | GoTaq Green Master Mix | Promega | Cat# M7123 | |
| Commercial assay or kit | TOPO TA Cloning Kit | Thermo Fisher Scientific | Cat# K452001 | |
| Commercial assay or kit | Monarch Plasmid Miniprep Kit | NEB | Cat# T1010L | |
| Commercial assay or kit | MinElute PCR Purification Kit | QIAGEN | Cat# 28006 | |
| Commercial assay or kit | iScript cDNA Synthesis Kit | Bio-Rad | Cat# 1708890 | |
| Commercial assay or kit | iQ SYBR Green Supermix | Bio-Rad | Cat# 1708880 | |

*Continued on next page*

*Continued*

| Reagent type (species) or resource | Designation | Source or reference | Identifiers | Additional information |
|---|---|---|---|---|
| Commercial assay or kit | Human Stem Cell Nucleofector Kit 2 | Lonza | Cat# VPH-5022 | |
| Commercial assay or kit | XtremeGene 9 DNA Transfection Reagent | Sigma-Aldrich | Cat# 06365779001 | |
| Commercial assay or kit | Cytofix/Cytoperm W/Golgi Stop Kit | BD Biosciences | Cat# 554715 | |
| Commercial assay or kit | Insulin ELISA Jumbo | Alpco | Cat# 80-INSHU-E10.1 | |
| Commercial assay or kit | Pierce BCA Protein Assay Kit | Thermo Fisher Scientific | Cat# 23227 | |
| Commercial assay or kit | TnT Coupled Wheat Germ Extract System | Promega | Cat# L4130 | |
| Chemical compound, drug | Penicillin-Streptomycin | Thermo Fisher Scientific | Cat# 15140122 | |
| Chemical compound, drug | Puromycin dihydrochloride | Sigma-Aldrich | Cat# P8833 | |
| Chemical compound, drug | ALK5 Inhibitor II | Enzo Life Sciences | Cat# ALX-270–445 | |
| Chemical compound, drug | Retinoic Acid | Sigma-Aldrich | Cat# R2625 | |
| Chemical compound, drug | Ascorbic Acid | Sigma-Aldrich | Cat# A4403-100MG | |
| Chemical compound, drug | LDN-193189 | Stemgent | Cat# 04–0074 | |
| Chemical compound, drug | SANT-1 | Sigma-Aldrich | Cat# S4572 | |
| Chemical compound, drug | TPB | Calbiochem | Cat# 565740 | |
| Chemical compound, drug | TGFβ R1 kinase inhibitor IV | EMD Biosciences | Cat# 616454 | |
| Chemical compound, drug | KAAD-Cyclopamine | Toronto Research Chemicals | Cat# K171000 | |
| Chemical compound, drug | TTNPB | Sigma-Aldrich | Cat# T3757 | |
| Chemical compound, drug | Cycloheximide | Sigma-Aldrich | Cat# C7698 | |
| Chemical compound, drug | Actinomycin D | Sigma-Aldrich | Cat# A9415 | |

*Continued on next page*

*Continued*

| Reagent type (species) or resource | Designation | Source or reference | Identifiers | Additional information |
|---|---|---|---|---|
| Chemical compound, drug | Polyethylenimine (PEI) | Polysciences | Cat# 23966–1 | |
| Chemical compound, drug | Hoechst 33342, Trihydrochloride, Trihydrate | Thermo Fisher Scientific | Cat# H3570 | |
| Chemical compound, drug | MitoSOX Red | Thermo Fisher Scientific | Cat# M36008 | |
| Chemical compound, drug | D-(+)-Glucose Solution, 45% | Sigma-Aldrich | Cat# G8769 | |
| Chemical compound, drug | Sodium Bicarbonate | Sigma-Aldrich | Cat# NC0564699 | |
| Chemical compound, drug | ROCK Inhibitor Y-27632 | STEMCELL Technologies | Cat# 72305 | |
| Software, algorithm | Flowjo-v10 | FlowJo, LLC | RRID:SCR_008520 | http://www.flowjo.com/download-newest-version/ |
| Software, algorithm | STAR 2.5.3a | *Dobin et al., 2013* | RRID:SCR_015899 | https://github.com/alexdobin/STAR |
| Software, algorithm | Bowtie 1.1.1 | *Langmead et al., 2009* | RRID:SCR_005476 | http://bowtie-bio.sourceforge.net/index.shtml |
| Software, algorithm | Cufflinks 2.2.1 | *Trapnell et al., 2010* | RRID:SCR_014597 | https://github.com/cole-trapnell-lab/cufflinks |
| Software, algorithm | HTSeq 0.6.1 | *Anders et al., 2015* | RRID:SCR_005514 | https://htseq.readthedocs.io/en/master/install.html |
| Software, algorithm | DEseq2 1.10.1 | *Love et al., 2014* | RRID:SCR_015687 | https://www.bioconductor.org/packages/devel/bioc/html/DESeq2.html |
| Software, algorithm | RiboTaper | *Calviello et al., 2016* | RRID:SCR_018880 | https://ohlerlab.mdc-berlin.de/software/ |
| Software, algorithm | R 3.5.0 | | RRID:SCR_001905 | https://cran.r-project.org/ |
| Software, algorithm | SAMtools 1.3 | *Li et al., 2009* | RRID:SCR_002105 | https://github.com/samtools/samtools |
| Software, algorithm | BEDTools 2.17.0 | *Quinlan and Hall, 2010* | RRID:SCR_006646 | https://bedtools.readthedocs.io/en/latest/content/installation.html |
| Software, algorithm | HOMER 4.10 | *Heinz et al., 2010* | RRID:SCR_010881 | http://homer.ucsd.edu/homer/download.html |
| Software, algorithm | GREAT 3.0.0 | *McLean et al., 2010* | RRID:SCR_005807 | http://great.stanford.edu/public/html/ |

*Continued on next page*

*Continued*

| Reagent type (species) or resource | Designation | Source or reference | Identifiers | Additional information |
|---|---|---|---|---|
| Software, algorithm | Adobe Illustrator CS5 | Adobe | RRID:SCR_010279 | |
| Software, algorithm | Adobe Photoshop CS5 | Adobe | RRID:SCR_014199 | |
| Software, algorithm | GraphPad Prism v7.05 | GraphPad Software, LLC | RRID:SCR_002798 | |
| Other | Novex 16% Tricine Protein Gel | Thermo Fisher Scientific | Cat# EC66955BOX | |
| Other | Novex Tricine SDS Sample Buffer (2X) | Thermo Fisher Scientific | Cat# LC1676 | |
| Other | Immobilon-PSQ PVDF Membrane | Merck Millipore | Cat# ISEQ00010 | |
| Other | Stellaris RNA FISH Hybridisation Buffer | LGC Biosearch Technologies | Cat# SMF-HB1-10 | |
| Other | Stellaris RNA FISH Wash Buffer A | LGC Biosearch Technologies | Cat# SMF-WA1-60 | |
| Other | Stellaris RNA FISH Wash Buffer B | LGC Biosearch Technologies | Cat# SMF-WB1-20 | |
| Other | QuickExtract DNA Extraction Solution | Lucigen | Cat# QE09050 | |
| Other | Vectashield Antifade Mounting Medium | Vector Laboratories | Cat# H-1000 | |
| Other | FastDigest BpiI | Thermo Fisher Scientific | Cat# FD1014 | |
| Other | FastDigest BshTI | Thermo Fisher Scientific | Cat# FERFD1464 | |
| Other | FastDigest SalI | Thermo Fisher Scientific | Cat# FD0644 | |
| Other | TRIzol | Thermo Fisher Scientific | Cat# 15596018 | |
| Other | Matrigel | Corning | Cat# 356231 | |
| Other | GlutaMAX | Thermo Fisher Scientific | Cat# 35050061 | |
| Other | DPBS (without calcium and magnesium) | Corning | Cat# 45000–434 | |
| Other | mTeSR1 Complete Kit - GMP | STEMCELL Technologies | Cat# 85850 | |
| Other | RPMI 1640 Medium, HEPES | Thermo Fisher Scientific | Cat# 22400–089 | |
| Other | DMEM/F12 with L-Glutamine, HEPES | Corning | Cat# 45000–350 | |

*Continued on next page*

*Continued*

| Reagent type (species) or resource | Designation | Source or reference | Identifiers | Additional information |
|---|---|---|---|---|
| Other | Dulbecco's Modified Eagle's Medium | Corning | Cat# 45000–312 | |
| Other | HyClone Dulbecco's Modified Eagles Medium | Thermo Fisher Scientific | Cat# SH30081.FS | |
| Other | MCDB 131 | Thermo Fisher Scientific | Cat# 10372–019 | |
| Other | Opti-MEM Reduced Serum Medium | Thermo Fisher Scientific | Cat# 31985062 | |
| Other | Insulin-Transferrin-Selenium-Ethanolamine (ITS-X) (100X) | Thermo Fisher Scientific | Cat# 51500–056 | |
| Other | B-27 Supplement (50X) | Thermo Fisher Scientific | Cat# 17504044 | |
| Other | Bovine Albumin Fraction V (7.5%) | Thermo Fisher Scientific | Cat# 15260037 | |
| Other | Fetal Bovine Serum | Corning | Cat# 35011CV | |
| Other | Donkey Serum | Gemini Bio-Products | Cat# 100-151/500 | |
| Other | Fatty Acid-Free BSA | Proliant Biologicals | Cat# 68700 | |
| Other | Accutase | eBioscience | Cat# 00-4555-56 | |

## HEK293T cell culture

HEK293T cells (female) were cultured in a humidified incubator at 37°C with 5% $CO_2$ using Dulbecco's Modified Eagle Medium (Corning; 4.5 g/L glucose, [+] L-glutamine, [-] sodium pyruvate) supplemented with 10% fetal bovine serum (FBS; Corning, Cat# 35011CV). HEK293T cells were purchased from ATCC (Cat# CRL-3216, RRID:CVCL_0063) and tested for mycoplasma prior to the experiment.

## hESC culture and maintenance

H1 hESCs (male) were obtained from WiCell (NIHhESC-10–0043, RRID:CVCL_9771) and tested for mycoplasma on a yearly basis. H1 hESCs were grown in feeder-independent conditions on Matrigel-coated dishes (Corning, Cat# 356231) with mTeSR1 media (STEMCELL Technologies, Cat# 85850). Propagation was carried out by passing the cells every 3 to 4 days using Accutase (eBioscience, Cat# 00-4555-56) for enzymatic cell dissociation. hESC research was approved by the University of California, San Diego, Institutional Review Board and Embryonic Stem Cell Research Oversight Committee.

## Pancreatic differentiation

H1 hESCs were differentiated in a monolayer format as previously described (*Rezania et al., 2012*), with minor modifications. Undifferentiated hESCs were seeded into 24-wells at $0.4 \times 10^6$ cells/well in 500 µl mTeSR1 medium. The next day the cells were washed in RPMI media (Thermo Fisher Scientific, Cat# 22400–089) and then differentiated with daily media changes. In addition to GlutaMAX (Thermo Fisher Scientific, Cat# 35050061), RPMI medium was supplemented with 0.12% (w/v) $NaHCO_3$ and 0.2% (Day 0) or 0.5% (Day 1–3) (v/v) FBS (Corning, Cat# 35011CV). DMEM/F12 medium (Corning, Cat# 45000–350) was supplemented with 2% (v/v) FBS and 0.2% (w/v) $NaHCO_3$, and DMEM High Glucose medium (HyClone, Thermo Fisher Scientific, Cat# SH30081.FS) was

supplemented with 0.5X B-27 supplement (Thermo Fisher Scientific, Cat# 17504044). Human Activin A, mouse Wnt3a, human KGF, and human Noggin were purchased from R and D Systems (Cat# 338-AC/CF, Cat# 1324-WN-010, Cat# 251 KG, Cat# 3344 NG-050). Other media components included TGFβ R1 kinase inhibitor IV (EMD Bioscience, Cat# 616454), KAAD-Cyclopamine (Toronto Research Chemicals, Cat# K171000), the retinoid analog TTNPB (Sigma Aldrich, Cat# T3757), the protein kinase C activator TPB (EMD Chemicals, Cat# 565740), the BMP type one receptor inhibitor LDN-193189 (Stemgent, Cat# 04–0074), and an inhibitor of the TGF-β type one activin like kinase receptor ALK5, ALK5 inhibitor II (Enzo Life Sciences, Cat# ALX-270–445).

Stage 1 (DE; collection on day 3):
Day 0: RPMI/FBS, 100 ng/mL Activin A, 25 ng/mL mouse Wnt3a
Day 1–2: RPMI/FBS, 100 ng/mL Activin A
Stage 2 (GT; collection on day 6):
Day 3: DMEM/F12/FBS, 2.5 µM TGFβ R1 kinase inhibitor IV, 50 ng/mL KGF
Day 4–5: DMEM/F12/FBS, 50 ng/mL KGF
Stage 3 (PP1; collection on day 10):
Day 6–9: DMEM/B27, 3 nM TTNPB, 0.25 mM KAAD-Cyclopamine, 50 ng/mL Noggin
Stage 4 (PP2; collection on day 13):
Day 10–12: DMEM/B27, 100 nM ALK5 inhibitor II, 100 nM LDN-193189, 500 nM TPB, 50 ng/mL Noggin
Stage 5 (endocrine cell stage; collection on day 16):
Day 13–15: DMEM/B27, 100 nM ALK5 inhibitor II, 100 nM LDN-193189, 500 nM TPB, 50 ng/mL Noggin

For ribosome profiling experiments, a scalable suspension culture protocol was employed for differentiation of H1 cells to the PP2 stage (*Rezania et al., 2014*). Undifferentiated hESCs were aggregated by preparing a single cell suspension in mTeSR1 media (STEMCELL Technologies; supplemented with 10 µM Y-27632) at $1 \times 10^6$ cells/mL and overnight culture in six-well ultra-low attachment plates (Costar) with 5.5 ml per well on an orbital rotator (Innova2000, New Brunswick Scientific) at 100 rpm. The following day, undifferentiated aggregates were washed in MCDB 131 media (Thermo Fisher Scientific, Cat# 10372–019) and then differentiated using a multistep protocol with daily media changes and continued orbital rotation at either 100 rpm or at 115 rpm from days 8 to 14. In addition to 1% GlutaMAX (Gibco, Thermo Fisher Scientific, Cat# 35050061) and 10 mM (days 0–10) or 20 mM (days 11–14) glucose, MCDB 131 media was supplemented with 0.5% (days 0–5) or 2% (days 6–14) fatty acid-free BSA (Proliant Biologicals, Cat# 68700), 1.5 g/L (days 0–5 and days 11–14) or 2.5 g/L (days 6–10) NaHCO$_3$ (Sigma-Aldrich), and 0.25 mM ascorbic acid (days 3–10).

Human Activin A, mouse Wnt3a, and human KGF were purchased from R and D Systems (Cat# 338-AC/CF, Cat# 1324-WN-010, Cat# 251 KG). Other media components included Insulin-Transferrin-Selenium-Ethanolamine (ITS-X; Thermo Fisher Scientific, Cat# 51500–056; days 6–10), retinoic acid (RA) (Sigma-Aldrich, Cat# R2625), the sonic hedgehog pathway inhibitor SANT-1 (Sigma-Aldrich, Cat# S4572), the protein kinase C activator TPB (EMD Chemicals, Cat# 565740), the BMP type one receptor inhibitor LDN-193189 (Stemgent, Cat# 04–0074), and the TGFβ type one activin like kinase receptor ALK5 inhibitor, ALK5 inhibitor II (Enzo Life Sciences, Cat# ALX-270–445).

Stage 1 (DE; collection on day 3):
Day 0: MCDB 131, 100 ng/mL Activin, 25 ng/mL mouse Wnt3a
Day 1–2: MCDB 131, 100 ng/mL Activin A
Stage 2 (GT; collection on day 6):
Day 3 – Day 5: MCDB 131, 50 ng/mL KGF
Stage 3 (PP1; collection on day 8)
Day 6 – Day 7: MCDB 131, 50 ng/mL KGF, 0.25 µM SANT-1, 1 µM RA 100 nM LDN-193189, 200 nM TPB
Stage 4 (PP2; collection on day 11):
Day 8 – Day 10: MCDB 131, 2 ng/mL KGF, 0.25 µM SANT-1, 0.1 µM RA, 200 nM LDN-193189, 100 nM TPB

## CRISPR/Cas9-mediated lncRNA knockout

To generate clonal lncRNA knockout hESC lines, combinations of pSpCas9(BB)−2A-Puro plasmid pairs (Addgene plasmid #62988, RRID:Addgene_62988, gift from Feng Zhang) expressing Cas9 and

single sgRNAs targeting upstream and downstream regions of the lncRNA promoter/locus were co-transfected into $1.5 \times 10^6$ H1 hESCs using the Human Stem Cell Nucleofector Kit 2 (Lonza) and the Amaxa Nucleofector II (Lonza). 24 hr after plating into Matrigel-coated six-well plates, nucleofected cells were selected with puromycin (1 μg/mL mTeSR1 media) for 2–3 consecutive days. Individual colonies that emerged within 7 days after transfection were subsequently transferred manually into 96-well plates for expansion. Genomic DNA for PCR genotyping with GoTaq Green Mastermix (Promega) and Sanger sequencing was then extracted using QuickExtract DNA Extraction Solution (Lucigen).

To generate sORF frameshift mutations, sgRNA sequences targeting the N-terminal region of the predicted small peptides were inserted into pSpCas9(BB)−2A-GFP (Addgene plasmid #48138, RRID: Addgene_48138, gift from Feng Zhang) via its BpiI cloning sites. 3 μg of the resulting plasmids were then transfected into 500,000 H1 cells plated into Matrigel-coated six-wells the day prior, using XtremeGene 9 Transfection Reagent (Sigma-Aldrich) according to the manufacturer's instructions. 24 hr post-transfection, 10,000 GFP$^+$ cells were sorted on an Influx Cell Sorter (BD Biosciences) into Matrigel-coated six-wells containing 1 mL mTeSR1 media supplemented with 10 μM ROCK inhibitor and 1X penicillin/streptomycin. Seven days after sorting, emerging colonies were hand-picked and transferred into 96-well plates for genotyping. Frameshifts inside the targeted sORFs were confirmed by PCR-amplification of the sORF sequence with GoTaq Green Mastermix (Promega, Cat# M7123) and subsequent subcloning the PCR products into pCR2.1 (Thermo Fisher Scientific). For each hESC clone, at least six pCR2.1 clones were Sanger sequenced. Oligonucleotide sequences for sgRNA cloning are provided in *Figure 4—source data 2A*.

## PCR genotyping of CRISPR clones

Four days after transfer of single cell-derived clones into 96-wells, cell culture supernatants containing dead cells were collected from each well prior to the daily media change. Cell debris was then pelleted and used for gDNA extraction with 10–20 μl QuickExtract DNA Extraction Solution (Lucigen, Cat# QE09050) according to the manufacturer's instructions. 1 μl DNA was then PCR-amplified with GoTaq Green Mastermix (Promega, Cat# M7123) and locus-specific primers that anneal either within or outside of the excised genomic DNA. PCR products generated with 'inside' primers were visualized on a 2% agarose gel, PCR bands generated with primers flanking the deletion were gel-purified and submitted for Sanger sequencing (see *Figure 4—source data 2B* for genotyping and sequencing primers).

For genotyping of sORF frameshift clones, PCR amplicons designed to encompass the Cas9 cut site were amplified and Sanger sequenced (*Figure 4—source data 2B*). If out-of-frame indels were apparent in the sequencing chromatogram, the sequenced PCR product was ligated into pCR2.1-TOPO via TOPO-TA cloning. A minimum of six clones were Sanger sequenced in order to determine the genotype at both alleles with high confidence.

## Generation of sORF translation reporter plasmids

The four lncRNAs tested were PCR-amplified with KOD Xtreme DNA Hotstart Polymerase (Millipore) from their 5' end up until the last codon of the sORF to be tested, omitting its stop codon (primer sequences are listed in *Figure 4—source data 2D*). cDNA was used as PCR template for *LINC00261* and *LHFPL3-AS2; RP11-834C11.4*, and *MIR7-3HG* were amplified from a gBlock synthetic gene fragment (Integrated DNA Technologies; see *Figure 4—source data 2F*). The GFP coding sequence (without start codon; amplified from pRRLSIN.cPPT.PGK-GFP.WPRE; RRID:Addgene_12252) was then fused in-frame to the sORF via overlap extension PCR. The resulting fusion product was cloned into pRRLSIN.cPPT.PGK-GFP.WPRE (Addgene plasmid #12252, gift from Didier Trono) via BshTI and SalI restriction sites included in the PCR primers. Due to the 3'-location of sORF7 within *LINC00261*, not the entire *LINC00261* cDNA was amplified but only 65 bp preceding sORF7.

To create the *RP11-834C11.4*-sORF-1XFLAG reporter construct in an analogous way, a gBlock synthetic gene fragment encompassing the FLAG-tagged sORF served as PCR template (*Figure 4—source data 2F*). The resulting PCR product was cloned into pRRLSIN.cPPT.PGK-GFP.WPRE via BshTI and SalI restriction sites.

## Generation of *LINC00261* wild type and ΔATG<sup>sORF1-7</sup> expression plasmids

The *LINC00261* wild type cDNA was PCR-amplified from pENTR/D-TOPO-*LINC00261* (gift from Leo Kurian) with KOD Xtreme DNA Hotstart Polymerase (Millipore, Cat# 71975). The resulting PCR product was inserted into pRRLSIN.cPPT.PGK-GFP.WPRE via its appended BshTI/SalI cloning sites. Full-length *LINC00261* ΔATG<sup>sORF1-7</sup> was assembled through overlap extension PCR from the following three fragments and subsequently cloned into pRRLSIN.cPPT.PGK-GFP.WPRE via appended BshTI/SalI cloning sites: (i) a 1,248 bp PCR product amplified from a synthetic gene construct (Genewiz; see *Figure 4—source data 2F* for sequence) in which the ATG start codons of sORFs 1–6 had been mutated (ATG → AAG/ATT/ AGG/AAG/ ATA/AGG), and (ii-iii) 3,111 bp/610 bp PCR fragments (amplified from the *LINC00261* cDNA) in which the sORF7 start codon was mutated (ATG → AAG). The obtained plasmids were sequence-verified by Sanger sequencing.

## Immunofluorescence staining

H1 hESC-derived cells grown as monolayer on Matrigel-coated coverslips were washed twice with PBS and then fixed with 4% paraformaldehyde in PBS for 30 min at room temperature. Following three washes in PBS, samples on coverslips were permeabilized and blocked with Permeabilization/ Blocking Buffer (0.15% (v/v) Triton X-100% and 1% normal donkey serum in PBS) for 1 hr at room temperature. Primary and secondary antibodies were diluted in Permeabilization/Blocking Buffer. Sections were incubated overnight at 4°C with primary antibodies, and then secondary antibodies for 30 min at room temperature. The following primary antibodies were used: rabbit anti-OCT4 (Cell Signaling Technology, Cat# 2890, RRID:AB_2167725, 1:500), goat anti-SOX17 (Santa Cruz Biotechnology, Cat# AF1924, RRID:AB_355060, 1:250), goat anti-FOXA2 (Santa Cruz Biotechnology, Cat# sc-6554, RRID:AB_2262810, 1:250), goat anti-GATA6 (Santa Cruz Biotechnology, Cat# sc-9055, RRID:AB_2108768, 1:50), guinea pig anti-insulin (Dako, Cat# A0564, RRID:AB_10013624). Secondary antibodies (1:1000) were Cy3- or Alexafluor488-conjugated antibodies raised in donkey against guinea pig, rabbit or goat (Jackson ImmunoResearch Laboratories, Cat# 706-545-148, RRID:AB_2340472, Cat# 711-545-152, RRID:AB_2313584, Cat# 705-165-147, RRID:AB_2307351). Images were acquired on a Zeiss Axio-Observer-Z1 microscope with a Zeiss AxioCam digital camera, and figures prepared with Adobe Photoshop/Illustrator CS5.

## Flow cytometry analysis

For intracellular flow cytometry, single cells were washed three times in FACS buffer (0.1% (w/v) BSA (Thermo Fisher Scientific in PBS) and then fixed and permeabilized with Cytofix/Cytoperm Fixation/ Permeabilization Solution (BD Biosciences) for 20 min at 4°C, followed by two washes in BD Perm/ Wash Buffer. Cells were next incubated with either PE-conjugated anti-SOX17 antibody (BD Biosciences; Cat# 561591, RRID:AB_10717121), or PE-conjugated anti-INS antibody (Cell Signaling Technology, Cat# 8508S, RRID:AB_11179076) in 50 µl BD Perm/Wash Buffer for 60 min at 4°C. Following three washes in BD Perm/Wash Buffer, cells were analyzed on a FACSCanto II (BD Biosciences) cytometer.

## Insulin content measurements

To measure total insulin content of endocrine cell stage control and lncRNA KO cells, adherent cultures were enzymatically detached from a 24-well at day 16 of differentiation. Upon quenching with FACS buffer (0.1% (w/v) BSA (Thermo Fisher Scientific in PBS), the cells were pelleted and extracted over night at 4°C in 100 µl acid-ethanol (2% HCl in 80% ethanol). Insulin was measured by Insulin ELISA (Alpco, Cat# 80-INSHU-E10.1) and normalized to total protein, as quantified with a BCA protein assay (Thermo Fisher Scientific, Cat# 23227).

## Quantitative reverse transcription PCR (qRT-PCR)

Total RNA was isolated from hESC-derived cells and HEK293T cells using either TRIzol (Thermo Fisher Scientific, Cat# 15596018) or the RNAeasy Mini Kit (Qiagen, Cat# 15596018), respectively. Upon removal of genomic DNA (TURBO DNA-free Kit, Thermo Fisher Scientific, Cat# AM1907 or RNase-free DNase Set, Qiagen, Cat# 79254) cDNA was synthesized using the iScript cDNA Synthesis Kit (Bio-Rad, Cat# 1708890). PCR reactions were run in triplicate with 6.25–12.5 ng cDNA per

reaction using the CFX96 Real-Time PCR Detection System (BioRad). TATA-binding protein (TBP) was used as endogenous control to calculate relative gene expression using the ΔΔCt method. Primer sequences are provided in *Figure 4—source data 2C*.

## Transient transfection of HEK293T cells with polyethylenimine (PEI)

Two hours prior to transfection, fresh pre-warmed DMEM medium (Corning, Cat# 45000–312) was added to each well. Transfection mix was prepared by combining PEI (Polysciences Cat# 23966–1) and plasmid DNA (4:1 ratio; 4 µg PEI per 1 µg DNA) in Opti-MEM Reduced Serum Medium (Thermo Fisher Scientific, Cat# 31985062) followed by brief vortexing. After five minutes, the transfection complex was added dropwise to the cells.

## Lentivirus preparation and ectopic *LINC00261* expression

Lentiviral particles were prepared by co-transfecting HEK293T cells (using PEI) with the pCMVR8.74/pMD2.G helper plasmids (Addgene plasmids #22036/12259, RRID:Addgene_22036 and RRID:Addgene_12259, gift from Didier Trono) and with pRRLSIN.cPPT.PGK-GFP.WPRE transfer plasmid (RRID:Addgene_12252), in which the GFP ORF had been replaced with the 4.9 kb *LINC00261* cDNA. Virus-containing supernatant was collected for two consecutive days and concentrated by ultracentrifugation for 2 hr at 19,400 rpm using an Optima L-80 XP Ultracentrifuge (Beckman Coulter).

To express *LINC00261* (wild type) and *LINC00261* (ΔATG$^{sORF1-7}$) in HEK293T cells, the cells were plated in 6-well plates and transduced with lentivirus the following day. Two days post infection, the cells were passaged for RNA half-life measurements.

## *LINC00261* RNA half-life measurement

HEK293T cells transduced with either *LINC00261* (wild type) or *LINC00261* (ΔATG$^{sORF1-7}$) lentivirus were seeded in six 24-wells. 48 hr after plating, cells from one well were collected for RNA isolation as the '0 hr' time point. To the remaining five wells, 100 µl growth media supplemented with 10 µg/ml actinomycin D (Sigma-Aldrich Cat# A9415) were added to inhibit transcription. At 2, 4, 6, 8, and 9 hr following actinomycin D addition, samples were collected for RNA isolation. Total RNA was then reverse transcribed and analyzed by qPCR, where the abundance of each time point was calculated relative to the abundance at the 0 hr time point (ΔCt). The half-life was then determined by non-linear regression (One phase decay; GraphPad Prism).

## Single molecule RNA fluorescence in situ hybridization (smRNA FISH)

H1-derived PP2 stage cells (control and *LINC00261* KO) were cultured on Matrigel-coated 12 mm diameter coverslips in a 24-well plate. Following 10 min fixation in 1 mL Fixation Buffer (3.7% (v/v) formaldehyde in PBS) at room temperature, the cells were washed twice in PBS and subsequently permeabilized in 70% (v/v) ethanol for one hour at 4°C. Following a five minute wash in Stellaris RNA FISH Wash Buffer A (LGC Biosearch Technologies, Cat# SMF-WA1-60; 1:5 diluted concentrate, with 10% (v/v) formamide added), the coverslips were incubated in a humidified chamber at 37°C for 14 hr with probes diluted in Stellaris RNA FISH Hybridisation Buffer (LGC Biosearch Technologies, Cat# SMF-HB1-10; with 10% (v/v) formamide added) to 125 nM. After a 30 min wash at 37°C in Wash Buffer A, the cells were counter-stained with Hoechst 33342 (Thermo Fisher Scientific) for 15 min and washed in RNA FISH Wash Buffer B (LGC Biosearch Technologies, Cat# SMF-WB1-20) for 5 min at room temperature. The coverslips were mounted in Vectashield Mounting Medium (Vector Laboratories, Cat# H-1000) and imaged on a UltraView Vox Spinning Disk confocal microscope (PerkinElmer) using a 100X oil objective.

## In vitro transcription/translation of lncRNAs

Synthetic gene constructs containing complete transcript isoforms (including the predicted 5' and 3' UTR) of four translated lncRNAs (*RP11-834C11.4*, *LINC00261*, *MIR7-3HG*, and *LHFPL3-AS2*) were produced by Genewiz (constructs available upon request). Microproteins were translated in vitro from 0.5 µg linearized plasmid DNA using the TnT Coupled Wheat Germ Extract system (Promega; Cat# L4140) in the presence of 10 mCi/mL [$^{35}$S]-methionine (Hartmann Analytic) according to manufacturer's instructions. 5 µL lysate was denatured for 2 min at 85°C in 9.6 µL Novex Tricine SDS Sample Buffer (2X) (Thermo Fisher Scientific; Cat# LC1676) and 1.4 µL DTT (500 mM). Proteins

were separated on 16% Tricine gels (Thermo Fisher Scientific; Cat# EC66955BOX) for 1 hr at 50 V followed by 3.5 hr at 100 V and blotted on PVDF-membranes (Immobilon-PSQ Membrane, Merck Millipore; Cat# ISEQ00010). Incorporation of [$^{35}$S]-methionine into newly synthesized proteins enabled the detection of translation products by phosphor imaging (exposure time of 1 day).

### In vivo translation assays

Reporter plasmids were transfected into HEK293T cells using PEI, and 36 hr post transfection live cells were imaged on an EVOS Cell Imaging System (Thermo Fisher Scientific) equipped with a 20X objective. Additional constructs were generated that served as negative controls (no GFP fluorescence): 1) a *LINC00261*-sORF3-GFP construct with a single 'T' insertion inside sORF3, causing a frame-shift, 2) a *LINC00261*-sORF2-GFP construct with a stop codon preceding the GFP coding sequence, and 3) a *LINC00261*-sORF1-GFP construct with a frame-shift mutation within the GFP coding sequence.

### Stranded mRNA-seq library preparation for lncRNA KOs

Total RNA from PP2 cells differentiated with the *Rezania et al., 2012* protocol was isolated and DNase-treated using either TRIzol (Thermo Fisher Scientific), or the RNAeasy Mini kit (Qiagen) according to the manufacturer's instructions. RNA integrity (RIN > 8) was verified on the Agilent 2200 TapeStation (Agilent Technologies), and 400 ng RNA was used for multiplex library preparation with the KAPA mRNA HyperPrep Kit (Roche; Cat# KK8581). All libraries were evaluated on TapeStation High Sensitivity DNA ScreenTapes (Agilent Technologies; Cat# 5067–5584) and with the Qubit dsDNA High Sensitivity (Life Technologies; Cat# Q10212) assays for size distribution and concentration prior to pooling the multiplexed libraries for single-end 1 × 51 nt or 1 × 75 sequencing on the HiSeq 2500 or HiSeq 4000 System (Illumina). Libraries were sequenced to a depth of > 20M uniquely aligned reads.

### Cell fractionation and ribo-minus RNA-seq

H1 hESCs were differentiated to the PP2 stage with the *Rezania et al., 2012* protocol, then nuclear and cytosolic RNA was isolated with the Paris Kit (Thermo Fisher Scientific). Unfractionated total RNA was set aside as a control. All samples were DNaseI-treated prior to further processing (TURBO DNA-free Kit; Thermo Fisher Scientific). rRNA-depleted total RNA-seq libraries were prepared with TruSeq Stranded Total RNA Library Prep Gold (Illumina; Cat# 2002059), and sequencing was performed on a HiSeq4000 instrument.

### Alignment of lncRNA KO mRNA-seq samples and processing for gene expression analysis

Using the Spliced Transcripts Alignment to a Reference (STAR) aligner (STAR 2.5.3b; *Dobin et al., 2013*), sequence reads were mapped to the human genome (hg38/GRCh38) with the Ensembl 87 annotations in 2-pass mapping mode, allowing for up to six mismatches. Cufflinks (part of the Cufflinks version 2.2.1 suite; *Roberts et al., 2011*; *Trapnell et al., 2010*), was then used to quantify the abundance of each transcript cataloged in the Ensembl 87 annotations in reads per kilobase per million mapped reads (RPKM). For plotting expression values, a pseudocount of 1 was added to all RPKM values prior to log$_2$-transformation.

Genes with RPKM ≥ 1 across two replicates were deemed expressed. Differential gene expression was tested using the DESeq2 v1.10.1 Bioconductor package (*Love et al., 2014*) with default parameters. Input count files for DESeq2 were created with htseq-count from the HTSeq Python library (*Anders et al., 2015*). Genes with a > 2 fold change and an adjusted p-value of <0.01 were considered differentially expressed.

The chromosomal localization of genes differentially expressed upon *LINC00261* KO was visualized with the RCircos package in R (https://cran.r-project.org/web/packages/RCircos/index.html).

### LncRNA classifications

The following transcript biotypes were grouped into the 'lncRNA' classification: 3' overlapping ncrna, antisense, bidirectional promoter lncRNA, lincRNA, macro lncRNA, non coding, processed transcript, sense intronic, sense overlapping, TEC.

LncRNAs with $\geq$1 RPKM during all differentiation stages of CyT49 hESCs (ES, DE, FG, GT, PP1, PP2) were categorized as constitutively expressed ('constitutive'), whereas lncRNAs with <1 RPKM throughout differentiation were considered 'never expressed'. LncRNAs expressed in at least one of the stages (but not in all five stages) were referred to as dynamically expressed ('dynamic'). Furthermore, for each lncRNA, its maximum RPKM value was determined across 38 tissues/cell types (see 'Gene-gene correlations and GO enrichment' section below). Log$_2$-transformed maximum expression values (RPKM + pseudocount of 1) were graphed as boxplots for different gene sets using the ggplot2 R package (https://cran.r-project.org/web/packages/ggplot2/index.html).

To determine the subcellular localization of lncRNAs, first all lncRNAs expressed in the nuclear and/or cytosolic RNA fraction (RPKM $\geq$ 1 in two biological replicates) of H1-derived PP2 stage cells were selected. Among these expressed lncRNAs, those with $\geq$ 1 RPKM$_{cytosol}$ and < 1 RPKM$_{nucleus}$ were classified as 'cytosol enriched'. Conversely, lncRNAs with < 1 RPKM$_{cytosol}$ and $\geq$ 1 RPKM$_{nucleus}$ were termed 'nucleus enriched'. LncRNAs expressed in both fractions ($\geq$1 RPKM$_{cytosol}$ and $\geq$ 1 RPKM$_{nucleus}$) were tagged with 'both'.

## Assignment of lncRNAs to their nearest coding gene using GREAT

GREAT (Genomic Regions Enrichment of Annotations Tool 3.0.0; *McLean et al., 2010*) was run with the 'Single nearest gene' within 1000 kb option to assign the nearest coding genes to the following sets of lncRNAs: i) DE-transcribed lncRNAs, ii) PP2-transcribed lncRNAs that are not transcribed at the DE stage (non-transcribed control set for i)), iii) PP2-transcribed lncRNAs, and iv) lncRNAs transcribed at the DE stage but not transcribed in PP2 cells (non-transcribed control set for iii)). The log$_2$-transformed RPKM values of the lncRNA-associated coding genes were then graphed as boxplots using ggplot2. The corresponding absolute coding-to-lncRNA inter-gene distances were visualized as cumulative frequency plots.

## Gene-gene correlations and GO enrichment

Pearson correlations were calculated among all genes across a catalog of 38 tissues/cell types derived from all three germ layers (16 Illumina BodyMap 2.0 tissues, other publicly available data sets (see 'Data sources' below), and EndoC-βH1 RNA-seq data generated in our lab). Scatter plots of the log$_2$-transformed RPKM values for lncRNAs/neighboring TFs and histograms of the Pearson correlation coefficients were plotted in R using ggplot2.

Spearman correlations were calculated to test for expression coregulation among all genes expressed (RPKM $\geq$ 1) in a minimum of ten out of 38 tissues. The resulting correlation matrix was then used to calculate the Euclidean distance followed by hierarchical clustering. The resulting heatmap was subdivided into ten clusters. Cluster visualization was done using heatmap.3 (https://raw.githubusercontent.com/obigriffith/biostar-tutorials/master/Heatmaps/heatmap.3.R) from gplots v3.0.1 (http://cran.r-project.org/web/packages/gplots/index.html). GO enrichment (*Ashburner et al., 2000*; *The Gene Ontology Consortium, 2019*) and KEGG pathway (*Kanehisa et al., 2017*) analyses to assign functional annotation to all ten clusters were performed with gProfiler v0.6.4 (*Reimand et al., 2016*) using g:Profiler archive revision 1741 (Ensembl 90, Ensembl Genomes 38).

## Alignment and processing of ChIP-seq samples

All sequence reads were filtered to include only those passing the standard Illumina quality filter, and then aligned to the *Homo sapiens* reference genome (hg38/GRCh38) using Bowtie version 1.1.1 (*Langmead et al., 2009*). The following parameters were used to select only uniquely aligning reads with a maximum of two mismatches:

$$-k1 - m1 - l50 - n2 - best - strata$$

SAMtools (*Li et al., 2009*) was then used to filter reads with a MAPQ score less than 30 and to remove duplicate reads. Finally, replicate ChIP-seq and input BAM files were merged and sorted. The HOMER makeUCSCfile function (*Heinz et al., 2010*) was used to create a bedGraph formatted file for viewing in the UCSC Genome Browser.

## Ribosome profiling and matching RNA-seq

Ribosome profiling was performed on PP2 cells obtained from six independent differentiations of H1 hESCs with the *Rezania et al., 2014* protocol, yielding an average of 89% PDX1-positive cells. Ribosome footprinting and sequencing library preparation was done with the TruSeq Ribo Profile (Mammalian) Library Prep Kit (Illumina, Cat# RPYSC12116, currently out of production) according to the TruSeq Ribo Profile (Mammalian) Reference Guide (version August 2016). In short, 50 mg of PP2 aggregates were washed twice with cold PBS and lysed for 10 min on ice in 1 mL lysis buffer (1 × TruSeq Ribo Profile mammalian polysome buffer, 1% Triton X-100, 0.1% NP-40, 1 mM dithiothreitol, 10 U ml$^{-1}$ DNase I, cycloheximide (0.1 mg/ml) and nuclease-free H$_2$O). Per sample, 400 µL of lysate was further processed according to manufacturer's instructions. Final library size distributions were checked on the Bioanalyzer 2100 using a High Sensitivity DNA assay (Agilent Technologies), multiplexed and sequenced on an Illumina HiSeq 4000 producing single end 1 × 51 nt reads. Ribo-seq libraries were sequenced to an average depth of 85M reads.

Total RNA was isolated using TRIzol Reagent (Thermo Fisher Scientific) from the exact same cell cultures processed for ribosome profiling (10% of the total number of cells). Total RNA was DNase treated and purified using the RNA Clean and Concentrator−25 kit (Zymo Research). RIN scores (RIN = 10 for all six samples) were measured on a BioAnalyzer 2100 using the RNA 6000 Nano assay (Agilent Technologies). Poly(A)-purified mRNA-seq library preparation was performed according to the TruSeq Stranded mRNA Reference Guide (Illumina), using 500 ng of total RNA as input. Libraries were multiplexed and sequenced on an Illumina HiSeq 4000 producing paired-end 2 × 101 nt reads.

## Alignment of Ribo-seq and matched mRNA-seq samples

Prior to mapping, ribosome-profiling reads were clipped for residual adapter sequences and filtered for mitochondrial, ribosomal RNA and tRNA sequences (*Figure 2—source data 1*). Next, all mRNA and ribosome profiling data were mapped to the Ensembl 87 transcriptome annotation of the human genome hg38 assembly using STAR 2.5.2b (*Dobin et al., 2013*) in 2-pass mapping mode. To avoid mRNA-seq mapping biases due to read length, the 2 × 101 nt mRNA-seq reads were next trimmed to 29-mers, and those mRNA reads were processed and mapped with the exact same settings as the ribosome profiling data. For the mapping of 2 × 101 nt RNA-seq reads six mismatches per read were allowed (default is 10), whereas two mismatches were permitted for the Ribo-seq and trimmed mRNA-seq reads. To account for variable ribosome footprint lengths, the search start point of the read was defined using the option `–seedSearchStartLmaxOverLread`, which was set to 0.5 (half the read, independent of ribosome footprint length). Furthermore, `–outFilterMultimapNmax` was set to 20 and `–outSAMmultNmax` to 1, which prevents the reporting of multimapping reads.

## Detecting actively translated reading frames

Canonical ORF detection using ribosome profiling data was performed with RiboTaper v1.3 (*Calviello et al., 2016*) with standard settings. For each sample, we selected only the ribosome footprint lengths for which at least 70% of the reads matched the primary ORF in a meta-gene analysis. Following the standard configuration of RiboTaper, we required ORFs to have a minimum length of 8aa, evidence from uniquely mapping reads and at least 21 P-sites. The final list of translation events was stringently filtered requiring the translated gene to have an average RNA RPKM $\geq$ 1 and to be detected as translated in all six profiled samples. Furthermore, we required the exact ORF to be detected independently in at least 4 out of 6 samples.

## Translational efficiency estimates

Translational efficiency (TE) estimations were calculated as the ratio of Ribo-seq over mRNA-seq DESeq2 normalized counts, yielding independent gene-specific TEs for each of the six individual replicate differentiations. For this, mRNA-seq and Ribo-seq based expression quantification was calculated for (annotated and newly detected) coding sequences (CDSs/ORFs) only, using RNA reads trimmed to footprint sizes as described above.

## Data sources

The following datasets used in this study were downloaded from the GEO and ArrayExpress repositories:

RNA-seq: Illumina BodyMap 2.0 expression data from 16 human tissues (GSE30611); polyA mRNA RNA-seq from BE2C (GSE93448), GM12878 (GSE33480), 293T (GSE34995), HeLa (GSE33480), HepG2 (GSE90322), HUVEC (GSE33480), Jurkat (GSE93435), K562 (GSE33480), Mia-PaCa-2 (GSE43770), Panc1 (GSE93450), PFSK-1 (GSE93451), U-87 MG (GSE90176); CyT49 hESC/DE/GT/PP1/PP2/CD142+ progenitors/CD200+ polyhormonal cells/in vivo matured endocrine cells/pancreatic islets (E-MTAB-1086).

ChIP-seq: H3K4me3/H3K27me3 in CyT49 hESC/DE/GT/PP1/PP2 (E-MTAB-1086).

## Statistical analyses

Statistical analyses were performed using Microsoft Excel, GraphPad Prism (7.05), and R (v.3.5.0).Statistical parameters such as the value of n, mean, standard deviation (S.D.), standard error of the mean (S.E.M.), significance level (*p<0.05, **p<0.01, ***p<0.001 and ****p<0.0001), and the statistical tests used are reported in the figures and figure legends. The ''n'' refers to the number of independent pancreatic differentiation experiments analyzed (biological replicates), or the number of genes/transcripts and sORFs detected.

Statistically significant gene expression changes were determined with DESeq2.

## Acknowledgements

We thank Andrea Carrano for comments on the manuscript and Francesca Mulas for advice with computational analyses. We acknowledge the UCSD IGM Genomics Center for next generation sequencing (P30 DK063491) and the UCSD Human Embryonic Stem Cell Core Facility for assistance with flow cytometry analysis and cell sorting. This work was supported by the National Institutes of Health (R01 DK068471 and R01 DK078803 to MS), an Alexander von Humboldt Foundation Research Award to MS, and a postdoctoral fellowship from the Larry L Hillblom Foundation (2015-D-021-FEL to BG). SvH was supported by an EMBO long-term fellowship (ALTF 186–2015, LTFCOFUND2013, GA-2013–609409). NH is the recipient of an ERC advanced grant under the European Union Horizon 2020 Research and Innovation Program (grant agreement AdG788970).

## Additional information

### Funding

| Funder | Grant reference number | Author |
| --- | --- | --- |
| National Institutes of Health | DK068471 | Maike Sander |
| Alexander von Humboldt-Stiftung | | Maike Sander |
| Larry L. Hillblom Foundation | 2015-D-021-FEL | Bjoern Gaertner |
| European Molecular Biology Organization | ALTF 186-2015 | Sebastiaan van Heesch |
| Horizon 2020 Framework Programme | AdG788970 | Norbert Hübner |
| National Institutes of Health | DK078803 | Maike Sander |

The funders had no role in study design, data collection and interpretation, or the decision to submit the work for publication.

### Author contributions

Bjoern Gaertner, Conceptualization, Data curation, Formal analysis, Investigation, Visualization, Methodology, Writing - original draft; Sebastiaan van Heesch, Formal analysis, Investigation, Visualization, Methodology, Writing - original draft; Valentin Schneider-Lunitz, Data curation, Formal analysis, Visualization; Jana Felicitas Schulz, Franziska Witte, Susanne Blachut, Steven Nguyen, Regina

Wong, Ileana Matta, Investigation, Methodology; Norbert Hübner, Conceptualization, Supervision, Funding acquisition, Project administration; Maike Sander, Conceptualization, Supervision, Funding acquisition, Investigation, Writing - original draft, Project administration

### Author ORCIDs
Sebastiaan van Heesch  https://orcid.org/0000-0001-9593-1980
Maike Sander  https://orcid.org/0000-0001-5308-7785

### Decision letter and Author response
Decision letter https://doi.org/10.7554/eLife.58659.sa1
Author response https://doi.org/10.7554/eLife.58659.sa2

## Additional files

### Supplementary files
• Transparent reporting form

### Data availability
All mRNA-seq and Ribo-seq datasets generated for this study have been deposited at GEO under the accession number GSE144682.

The following dataset was generated:

| Author(s) | Year | Dataset title | Dataset URL | Database and Identifier |
|---|---|---|---|---|
| Gaertner B, van Heesch S, Schneider-Lunitz V, Schulz JF, Witte F, Blachut S, Nguyen S, Wong R, Matta I, Hubner N, Sander M | 2020 | The role of long noncoding RNAs during pancreas development | http://www.ncbi.nlm.nih.gov/geo/query/acc.cgi?acc=GSE144682 | NCBI Gene Expression Omnibus, GSE144682 |

The following previously published datasets were used:

| Author(s) | Year | Dataset title | Dataset URL | Database and Identifier |
|---|---|---|---|---|
| Khrebtukova I | 2011 | Illumina BodyMap 2.0 | http://www.ncbi.nlm.nih.gov/geo/query/acc.cgi?acc=GSE30611 | NCBI Gene Expression Omnibus, GSE30611 |
| ENCODE project consortium | 2012 | RNA-seq from ENCODE/Caltech | http://www.ncbi.nlm.nih.gov/geo/query/acc.cgi?acc=GSE33480 | NCBI Gene Expression Omnibus, GSE33480 |
| ENCODE Project Consortium | 2012 | polyA mRNA RNA-seq from BE2C (ENCSR000BYK) | http://www.ncbi.nlm.nih.gov/geo/query/acc.cgi?acc=GSE93448 | NCBI Gene Expression Omnibus, GSE93448 |
| Huelga SC, Vu AQ, Arnold JD, Liang TY, Liu PP, Yan BY, Donohue JP, Shiue L, Hoon S, Brenner S, Ares M, Yeo GW | 2012 | Integrative genome-wide analysis reveals cooperative regulation of alternative splicing by hnRNP proteins (RNA-Seq) | http://www.ncbi.nlm.nih.gov/geo/query/acc.cgi?acc=GSE34995 | NCBI Gene Expression Omnibus, GSE34995 |
| ENCODE Project Consortium | 2016 | polyA mRNA RNA-seq from HepG2 (ENCSR329MHM) | http://www.ncbi.nlm.nih.gov/geo/query/acc.cgi?acc=GSE90322 | NCBI Gene Expression Omnibus, GSE90322 |
| ENCODE Project Consortium | 2017 | polyA mRNA RNA-seq from Jurkat clone E61 (ENCSR000BXX) | http://www.ncbi.nlm.nih.gov/geo/query/acc.cgi?acc=GSE93435 | NCBI Gene Expression Omnibus, GSE93435 |
| Sherman MH, Yu RT, Engle DD, | 2014 | Vitamin d receptor-mediated stromal reprogramming suppresses | http://www.ncbi.nlm.nih.gov/geo/query/acc.cgi? | NCBI Gene Expression Omnibus, |

| | | | | |
|---|---|---|---|---|
| Ding N, Atkins AR, Tiriac H, Collisson EA, Connor F, Van Dyke T, Kozlov S, Martin P, Tseng TW, Dawson DW, Donahue TR, Masamune A, Shimosegawa T, Apte MV, Wilson JS, Ng B, Lau SL, Gunton JE, Wahl GM, Hunter T, Drebin JA, O'Dwyer PJ, Liddle C, Tuveson DA, Downes M, Evans RM | | pancreatitis and enhances pancreatic cancer therapy | acc=GSE43770 | GSE43770 |
| ENCODE Project Consortium | 2017 | polyA mRNA RNA-seq from Panc1 (ENCSR000BYM) | http://www.ncbi.nlm.nih.gov/geo/query/acc.cgi?acc=GSE93450 | NCBI Gene Expression Omnibus, GSE93450 |
| ENCODE Project Consortium | 2017 | polyA mRNA RNA-seq from PFSK-1 (ENCSR000BYN) | http://www.ncbi.nlm.nih.gov/geo/query/acc.cgi?acc=GSE93451 | NCBI Gene Expression Omnibus, GSE93451 |
| ENCODE Project Consortium | 2016 | polyA mRNA RNA-seq from U-87 MG (ENCSR000BYO) | http://www.ncbi.nlm.nih.gov/geo/query/acc.cgi?acc=GSE90176 | NCBI Gene Expression Omnibus, GSE90176 |
| Xie R, Everett LJ, Lim HW, Patel NA, Schug J, Kroon E, Kelly OG, Wang A, D'Amour KA, Robins AJ, Won KJ, Kaestner KH, Sander M | 2013 | ChIP-seq and RNA-seq of coding RNA of the progression of human embryonic stem cells to beta cells to characterize the epigenetic programs that underlie pancreas differentiation | https://www.ebi.ac.uk/arrayexpress/experiments/E-MTAB-1086/ | ArrayExpress, E-MTAB-1086 |

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
