## [Decision Letter]

**Acceptance summary:**

The study nicely documents the existence of many cytoplasmic lncRNAs in the vicinity of lineage-determining transcription factors. Extensive functional analysis of one such lncRNA – *LINC00261* – demonstrates its importance in the differentiation of human embryonic stem cells into pancreatic endocrine cells. Interestingly, the function of *LINC00261* is independent of its potential microproteins and regulation of the neighboring transcription factor FOXA2. These studies highlight the complexity of lncRNA regulation of developmental processes.

**Decision letter after peer review:**

Thank you for submitting your article "A microprotein-independent role for highly translated lncRNA LINC00261 in pancreatic endocrine cell development" for consideration by *eLife*. Your article has been reviewed by three peer reviewers, and the evaluation has been overseen by Didier Stainier as the Senior Editor. The reviewers have opted to remain anonymous.

The reviewers have discussed the reviews with one another and the Reviewing Editor has drafted this decision to help you prepare a revised submission.

Summary:

This is a comprehensive study of mapping lncRNA expression during hESC differentiation towards the pancreatic endocrine cell lineage, combined with an evaluation of whether small open reading frames within each lncRNA could be expressed and have functional effects. Gaertner and co-workers map cytoplasm-located lncRNAS associated with translating ribosomes using the RiboSeq method and observed a comparatively frequent prediction of small open reading frames (smORFs) in these lncRNAs. Many of these smORFs could be translated in vitro using TnT assays. To investigate these smORF containing lncRNAs further, 10 were selected based on expression pattern and genomic context, and homozygous knock-out hESC lines were generated. One of these, LINC00261, was investigated in depth and each of its 7 smORFs were deleted in individual hESC lines. Although full-length LINC00261 was shown to have a role in INS+ cell differentiation, individual deletion of its smORFs did not affect INS+ cell differentiation. Simultaneous deletion of all 7 ORFs did not affect transcript stability; however functional activity of the mutated lncRNA wasn't tested. The authors conclude that the action of LINC00261 on hESC differentiation to INS+ cells is independent of the harbored smORFs.

Essential revisions:

Although this is an interesting study with novel information regarding lncRNAs in islet cell differentiation, the authors make claims throughout the manuscript that are not supported by their data, therefore their conclusions should be limited to what they actually show. The majority of these issues can be corrected by careful editing, but they also somewhat diminish the impact of the study. In addition, the manuscript text should be carefully checked for overstatements. For example, in the first paragraph of the Results section, the authors state that majority of lncRNAs showed dynamic expression patterns. These are not a majority as they represent 37%.

Primary concern: If the authors wish to claim that the linc00261 function is independent of the smORFs ability to be translated, then they must present data on differentiation capability cells having the simultaneously mutated smORFs, in addition to the linc00261 transcript stability data. In the current version, they have not probed the functional effects of mutating all the smORFs, which could be redundant. However, if they are not able to produce this data, then they will have to change the title and conclusions of the manuscript to reflect what they actually show.

As for their data on the smORFs (Figure 2), they have shown that the smORF-coding nucleotides bind to the ribosome and generate the classic 3-letter foot-print indicative of translation. They have shown that RNAs corresponding to the smORFs can undergo forced expression with translation into fusion protein using high expression levels in HEK293 cells. However, showing that the smORFs associate with the ribosome or can forcefully be translated, does not demonstrate that they are present under the conditions in which they investigate the function of linc00261 (in the ESC model of pancreatic differentiation). The authors could consider, if the smORFs could inhibit ribosome progression or release. Also, the peptides could be unstable and therefore not present endogenously. They must make proper limitations to their conclusions.

Furthermore, the authors claim that mutating all translated ORFs significantly reduce LINC00261 ability to bind ribosomes. They need to perform ribosome association studies with the mutated transcript to make this claim.

Although Figure 4H shows that sORF4 and sORF7 mutations in one of two clones causes a significant reduction in the number of insulin+ cells, the text claims there is no effect on deleting the ORFs. The authors need to resolve this discrepancy. What is the difference between the clones – if it was KO efficiency then it appears that deletion of these ORFs does have a phenotype.

It is surprising that disruption of the LINC00261 transcript has not phenotype and does not affect FOXA2 gene expression which contradicts previous published studies. At the very least, the authors need to acknowledge this and include some explanation in their Discussion.

Authors should add in a limitations section to list the study limitations such as small number of repeats, need for future experiments relating to single cell analyses and discuss why other reports in human datasets do not see any association in lnc00261 and islet beta cell function.

---

## [Author Response]

Essential revisions:Although this is an interesting study with novel information regarding lncRNAs in islet cell differentiation, the authors make claims throughout the manuscript that are not supported by their data, therefore their conclusions should be limited to what they actually show. The majority of these issues can be corrected by careful editing, but they also somewhat diminish the impact of the study. In addition, the manuscript text should be carefully checked for overstatements. For example, in the first paragraph of the Results section, the authors state that majority of lncRNAs showed dynamic expression patterns. These are not a majority as they represent 37%.

We thank the reviewers for their careful and thorough evaluation of our manuscript. As per their suggestion, we have reassessed all claims and statements made based on our experiments and computational analyses. We have adjusted claims where necessary. With respect to the example given (37% is not the majority), the given Results section reads:

"While some lncRNAs were constitutively expressed (n = 592; 25.3%), the majority showed dynamic expression patterns, being either strongly enriched in (n = 874; 37.4%), or specific to (n = 871; 37.3%) a single developmental intermediate of pancreatic lineage progression (Figure 1B and Figure 1—source data 1A)."

With the majority of dynamically expressed lncRNAs, we here referred to the sum of the stage-enriched (i.e. expressed in one or two stages, whilst absent in other stages) and stage-specific (i.e. expressed in a single evaluated stage only) lncRNAs. Together, these two groups of dynamically expressed lncRNAs sum up to a majority of 74.7% (37.4% + 37.3%) of all expressed lncRNAs. We apologize for this ambiguity in our writing and have adjusted this for better clarity in the revised version of the manuscript.

Primary concern: If the authors wish to claim that the linc00261 function is independent of the smORFs ability to be translated, then they must present data on differentiation capability cells having the simultaneously mutated smORFs, in addition to the linc00261 transcript stability data. In the current version, they have not probed the functional effects of mutating all the smORFs, which could be redundant. However, if they are not able to produce this data, then they will have to change the title and conclusions of the manuscript to reflect what they actually show.

We understand the reviewers' concern that differentiation of hESCs with all ORFs mutated simultaneously would provide a more definitive answer to the potential combined (or redundant) functions of the translated short ORFs and the microproteins produced from these short ORFs. However, given the size of the LINC00261 locus, introducing all seven point mutations simultaneously (or consecutively) into hESCs to generate ∆ATG mutant LINC00261 hESC lines is technically challenging, if not impossible. Such an experiment comes with significant caveats, such as off-target mutations due to repeated targeting and increased clone-to-clone variability caused by extensive passaging of the hESCs. In addition, due to the requirement for numerous rounds of gene editing, we would not be able to conduct these experiments within a reasonable timeframe for revision, especially now that our laboratory is operating at very limited capacity due to COVID-19.

Based on the experimental evidence we provide, what we can state regarding LINC00261 function is the following: (i) deletion of the entire transcript impairs endocrine cell differentiation and reduces insulin content in endocrine stage cultures, (ii) deletion of the entire transcript has no effect on FOXA2 expression, (iii) mutation of the individual ORFs (one-by-one) does not recapitulate this phenotype, and (iv) upon expression in HEK293T cells, LINC00261 transcripts with or without *all* ORFs mutated at once are equally stable – the evidence being that there is no difference in LINC00261 abundance in the presence of the transcription inhibitor actinomycin D (Figure 4K). In support of the latter observation, mutation of each of the single ORFs individually also had no effect on LINC00261 abundance in hESC-derived pancreatic cells (Figures 4J and Figure 4—figure supplement 1E). We agree with the reviewers that based on these observations we cannot rule out functional redundancy between the translated ORFs, as upon KO of a single ORF another (non-targeted) ORF or microprotein produced from this ORF may take over part of its functionality.

Our findings allow for the conclusion that the ORFs present in LINC00261 are not translated merely as a means to induce translation-coupled LINC00261 nonsense-mediated decay (NMD), a mechanism that was previously proposed as a potential main reason for lncRNA translation (Tani et al., 2013; Carlevaro-Fita et al., 2016). Taking away all endogenously favored translation initiation sites (at the very minimum) should strongly reduce the ability of the mutated transcript to attract ribosomes, though we show deleting these ORFs had no effect on LINC00261 stability.

As suggested by the reviewers, we have limited our claims to reflect the experimental data provided, and have adjusted the title and conclusions accordingly. We have added discussion on this topic (i.e. the absence of a mutant that has all ORFs mutated) and also refer to this point in the newly introduced 'limitations and future directions' paragraph.

The new title is: “A human ESC-based screen identifies a role for the translated lncRNA LINC00261 in pancreatic endocrine differentiation”

In the Discussion we state: “It would have been interesting to determine how concurrent mutation of all sORFs in LINC00261 affects pancreatic cell differentiation. However, given the size of the LINC00261 locus and the many sORFs, such an approach comes with technical challenges and significant caveats.”

In the limitations and future directions section we state: “LINC00261's microproteins and sORFs may have redundant functions which could explain why deletion of individual sORFs produces no apparent phenotype. Thus, despite limited sequence similarity and stark differences in translation rates between the identified translated sORFs in LINC00261, we cannot rule out that different microproteins produced by LINC00261 compensate when one sORF is deleted.”

As for their data on the smORFs (Figure 2), they have shown that the smORF-coding nucleotides bind to the ribosome and generate the classic 3-letter foot-print indicative of translation. They have shown that RNAs corresponding to the smORFs can undergo forced expression with translation into fusion protein using high expression levels in HEK293 cells. However, showing that the smORFs associate with the ribosome or can forcefully be translated, does not demonstrate that they are present under the conditions in which they investigate the function of linc00261 (in the ESC model of pancreatic differentiation). The authors could consider, if the smORFs could inhibit ribosome progression or release. Also, the peptides could be unstable and therefore not present endogenously. They must make proper limitations to their conclusions.

Whether all translated 'non-canonical' short ORFs (e.g. uORFs and lncRNA ORFs) produce stable proteins has been a subject of debate over the past years. Although we and others have systematically evaluated the protein-coding capacity and functional (regulatory) roles of many of these short ORFs (e.g. van Heesch et al., Cell 2019; Ji et al., Science 2020; Martinez et al., Nat Chem Biol 2019; Prensner et al., 2020), none of these studies provide an ultimate, overall classification that separates regulatory from protein producing ORFs, as these categories are per se not mutually exclusive. Furthermore, many of these ORFs may not be functional at all, but merely byproducts of evolutionary processes that have created the right circumstances for translation. What we do know from the aforementioned studies, is that for about 50-60% of all translated short ORFs microprotein products can generally be detected in vitro and in vivo (as determined by overexpression of lncRNAs with GFP/HA/V5/FLAG tagged ORFs, in vitro translation assays, in vivo shotgun and targeted mass-spectrometry (SRM/PRM), or HLA immunopeptidomics). Despite the challenges that come with the detection of such short protein products, these findings strongly suggest that many microproteins are stable enough to be detected by proteomics in human tissues and cell lines.

To elaborate on this a bit further, and to put our findings into a broader perspective, we investigated (i) how frequently the ORFs we annotate in this study have additionally been found independently in other human studies that used Ribo-seq to annotate short ORFs in lncRNAs, and (ii) for how many of these ORFs protein-level evidence for the produced microprotein could be found – either from shotgun, targeted, or immunopeptidomic (HLA bound) mass spectrometry approaches. In summary, of the 625 lncRNA ORFs (within 285 lncRNA genes) found to be translated in pancreatic progenitor cells, 410 (66%) are also identified as translated in one or more additional study (but not yet annotated by GENCODE, RefSeq or Ensembl), supporting the quality and reproducibility of the reported Ribo-seq-based ORF annotations. Of these 410 short ORFs, mass spec evidence presented in these additional studies could be found for 165 sORF-encoded microproteins (40% of all theoretically detectable ORFs). These results of course exclude short ORFs translated exclusively in our system (e.g. LINC00261, LHFPL3-AS2 or MIR7-3HG).

Importantly – and similar to the translation of many uORFs or alternative CDSs in mRNAs (e.g. NMD transcripts) – the fact that we can find protein products does not exclude a regulatory role for the act of translation of these ORFs. The importance of properly dissecting coding and noncoding roles of a single locus is one of the key points we attempted to make with our manuscript. Guided by gene biotypes, scientists have long and unknowingly ignored the likely dual coding and noncoding functions many genes can have, whether biotyped as lncRNA or protein coding. However, to properly investigate the function of a particular locus, potential coding or noncoding functions must be fully considered. For example, when deleting a protein product, one should ideally leave the RNA sequence and structure largely intact (e.g. by changing only the CDS's start codon, but not the nucleotide sequence and structure of the rest of the RNA). We have attempted to do this to the best of our capabilities for LINC00261, by complementing the knockout of the complete gene (RNA and ORFs) with ORF-by-ORF deletions that each leave the expression and sequence of LINC00261 intact.

As suggested by the reviewers, we have expanded the interpretation of our findings by discussing that the act of LINC00261 translation may additionally serve a regulatory purpose and by stressing that we have not evaluated protein-level evidence of LINC00261 microprotein production directly in the hESC differentiation system. We have specifically made note of this in the newly introduced 'limitations and future directions' paragraph.

In the limitations and future directions section we state: “Second, although we provide evidence that LINC00261 can produce microproteins using Ribo-seq, which is further supported by in vitro translation assays and overexpression of LINC00261 constructs with different in-frame tags, we provide no protein-level evidence for the endogenous production and stability of LINC00261's microproteins in this differentiation system or in human pancreas development in vivo.”

Furthermore, the authors claim that mutating all translated ORFs significantly reduce LINC00261 ability to bind ribosomes. They need to perform ribosome association studies with the mutated transcript to make this claim.

We agree that despite mutating multiple ORFs simultaneously, alternative ribosome binding at LINC00261 upon overexpression in HEK293T cells could theoretically occur. However, in the pancreatic progenitor cell Ribo-seq data used to annotate these ORFs, we see very little ribosome binding patterns outside of the newly annotated LINC00261 ORFs. In theory, upon ORF mutation new initiation sites could be selected not utilized under wild type conditions.

Since we provide no direct readout of altered ribosome binding upon ORF mutation, we have softened claims regarding absolute absence of ribosome binding in the revised version of the manuscript, and rephrased these claims to better represent what we have actually tested, i.e. that the absence of the translation of the ORFs endogenously translated in pancreatic progenitors does not influence the stability of LINC00261.

Although Figure 4H shows that sORF4 and sORF7 mutations in one of two clones causes a significant reduction in the number of insulin+ cells, the text claims there is no effect on deleting the ORFs. The authors need to resolve this discrepancy. What is the difference between the clones – if it was KO efficiency then it appears that deletion of these ORFs does have a phenotype.

Although we were initially excited to see a possible effect of sORF4 and sORF7 deletion, this effect could only be replicated in one of the two independently generated hESC KO clones. For each of the two clones (Figure 4H) between four and seven independent differentiations were conducted for these measurements, each in comparison to an equal number of wild type clone differentiations carried out side-by-side under the same culture conditions. Furthermore, measurement of insulin content in the same sORF4 and sORF7 KO endocrine cell stage cultures revealed no significant difference (Figure 4I), which contrasts our findings in LINC00261 KO clones (Figure 4C). The lack of a reproducible result between both clones and the lack of an effect on insulin production was sufficient reason for us to classify the observation as non-consistent, and we have now clarified this in the Results section of the revised manuscript.

All clones contain homozygous mutations for which individual colonies were handpicked and subsequently genotyped. Since for each clone, the ORF KO was validated by genotyping (through Sanger sequencing), seeing a phenotype in only a single clone is more likely to be the result of an unknown off target effect, rather than it being attributable to the (efficiency of the) ORF deletion itself.

In the Results we now state: “We observed no difference between sORF loss-of-function and control hESC lines for most of these endpoints (Figure 4H, I and Figure 4—figure supplement 1E), although we noticed that the number of insulin^+^ cells, but not the amount of insulin produced, was reduced in one of the two sORF4 and sORF7 KO clones.”

It is surprising that disruption of the LINC00261 transcript has not phenotype and does not affect FOXA2 gene expression which contradicts previous published studies. At the very least, the authors need to acknowledge this and include some explanation in their Discussion.

We were also surprised by this finding and the fact that LINC00261, as well as several other lncRNAs we deleted, did not affect the expression of neighboring (sense) protein-coding genes. However, as summarized below, the evidence that we collected is rigorous, and we are certain that our results are sound and correct, even though they contrast a number of previous studies that have reported a cis-regulatory relationship between LINC00261 and FOXA2 (e.g. Jiang et al., 2015; Swarr et al., 2019; Amaral et al., 2018). Differences in cell context and experimental approach could account for the differences, as detailed below.

We examined FOXA2 mRNA expression in LINC000261 knockout cells (LINC00261^-/-^ cells) at two stages of in vitro differentiation: (i) in definitive endoderm, which is the same cell context in which Jiang et al., 2015 observed FOXA2 regulation, and (ii) in pancreatic progenitors. We used qRT-PCR and RNA-seq as independent methods and both show no evidence for regulation of FOXA2 by LINC00261 (Figures 3D, Figure 3—figure supplement 1C, 4F, Figure 4—figure supplement 1C). In addition, we show that there is no difference in FOXA2 protein expression between control and LINC00261^-/-^ cells by immunofluorescence staining in definitive endoderm (Figure 3F). Thus, we provide clear and rigorous evidence in two distinct cellular contexts that LINC00261 does not regulate FOXA2 during human endoderm development.

One important difference between our study and the study by Jiang et al. is that we used a CRISPR-Cas9-mediated deletion strategy while Jiang et al. employed shRNAs to knockdown LINC00261 in human definitive endoderm. It is possible that shRNA-mediated knockdown produces different effects. Furthermore, compared to our differentiation protocol, the differentiation strategy applied by Jiang et al. to generate definitive endoderm was less efficient in inducing definitive endoderm. It is possible that cells generated with the Jiang et al. protocol may be inherently less stable in their cell fate choice, which could influence effects of LINC00261 loss-of-function on gene regulation. Amaral et al. studied LINC00261 in human cancer cell lines and similar to Jiang et al. used siRNAs rather than deletion. Swarr et al. studied LINC00261 by knockout in mice but not in human cells. While LINC00261 is positionally conserved between mouse and human, the sequence conservation is poor. Species differences might explain the difference between our findings in human cells and Swarr’s findings in mouse lung.

As suggested by the reviewers, we have included a detailed discussion of the discrepancies between our results and previous studies in the Discussion section under the heading “Limited cis-regulatory consequences of lncRNA deletion” of our revised manuscript.

Authors should add in a limitations section to list the study limitations such as small number of repeats, need for future experiments relating to single cell analyses and discuss why other reports in human datasets do not see any association in lnc00261 and islet beta cell function.

To address the concerns, we have expanded the Discussion and also added a limitations and future directions section to the Discussion, where we highlight the following aspects:

i) differences exist between differentiation protocols for the generation of definitive endoderm and endocrine cells. How much these recapitulate the in vivo situation is not known, but culture conditions could impact gene regulation and might explain the discrepancies between studies using different protocols. For example, the effect of LINC00261 knockout or knockdown on FOXA2 expression could be protocol dependent. We make clear that we use a stem-cell based in vitro differentiation system, and that we cannot make immediate inferences on conditions that govern human in vivo development.

ii) ORFs or the produced microproteins may have redundant functions, which we did not assess in our differentiation system (as we only target individual ORFs on a one-by-one basis).

iii) For the cytoplasmic and nuclear fractionation experiments, we have only used an n of two. We have adjusted the visualization of these data to clarify this and have mentioned this as a limitation to the interpretation of these data. We have been fully transparent on the precise number of replicates used for all experiments, throughout the figure legends, Results, and Materials and methods sections. We note that our findings on the role of LINC00261 in endocrine cell differentiation are supported by analysis of a total of *24* independent in vitro differentiations across four independently targeted hESC clones. Therefore, these findings have no limitations regarding replicate numbers.